# A Unified Stability Analysis of SAM vs SGD: Role of Data Coherence and Emergence of Simplicity Bias

**Wei-Kai Chang**
Purdue University
chang986@purdue.edu

**Rajiv Khanna**
Purdue University
rajivak@purdue.edu

## Abstract

Understanding the dynamics of optimization in deep learning is increasingly important as models scale. While stochastic gradient descent (SGD) and its variants reliably find solutions that generalize well, the mechanisms driving this generalization remain unclear. Notably, these algorithms often prefer flatter or simpler minima—particularly in overparameterized settings. Prior work has linked flatness to generalization, and methods like Sharpness-Aware Minimization (SAM) explicitly encourage flatness, but a unified theory connecting data structure, optimization dynamics, and the nature of learned solutions is still lacking. In this work, we develop a linear stability framework that analyzes the behavior of SGD, random perturbations, and SAM—particularly in two-layer ReLU networks. Central to our analysis is a coherence measure that quantifies how gradient curvature aligns across data points, revealing why certain minima are stable and favored during training. (Code are available in: `https://github.com/changwk1001/Stability_Analysis_and_Simplicity-Bias.git`)

## 1 Introduction

Modern deep networks often achieve low training error even in extreme overparameterized settings, yet they generalize surprisingly well. A key question is why the particular solutions found by standard training procedures tend to generalize, when many other parameter configurations could fit the training data but fail on test data. A body of work has suggested that stochastic gradient descent (SGD) implicitly favors solutions associated with *flat minima* (i.e. wide, low-curvature regions of the loss) which correlate with better generalization [Keskar et al., 2017, Hochreiter and Schmidhuber, 1997]. Recent algorithms like Sharpness-Aware Minimization (SAM) explicitly optimize for flatness and further improve generalization [Foret et al., 2021]. A complementary line of reasoning posits an implicit simplicity bias in overparameterized neural networks: given a choice of multiple functions that fit the training data, SGD tends to find those that rely on simpler or more "intuitive" features, rather than complex or idiosyncratic ones. Empirical evidence shows that neural networks often learn the most predictive yet simplest patterns in the data first, and may entirely ignore more complex features if the simple ones already suffice [Arpit et al., 2017, Shah et al., 2020]. This built-in Occam's razor has been offered as an explanation for why DNNs do not overfit even when they could in principle memorize the training set [Valle-Pérez et al., 2019].

Both the flat-minima hypothesis and the simplicity bias hypothesis provide important clues to neural network generalization. Yet it remains unclear how these perspectives connect, and what underlying mechanism drives this preferential selection of solutions by SGD. In particular, why should an algorithm like SGD prefer flat minima or feature-simple solutions in the first place? And how do modifications to the optimizer — such as adding random noise or using Sharpness-Aware Minimization (SAM) [Foret et al., 2021] — alter these preferences? Intriguingly, recent empirical evidence further suggests that even among multiple minima of equal overall flatness, SAM can

exhibit an additional implicit bias favoring those solutions that generalize better [Andriushchenko et al., 2023, Wen et al., 2023a, Springer et al., 2024]. This calls for a unified theoretical framework to explain which minima are favored by different training dynamics and why.

Towards this goal, we analyze the *linear stability* of different optimization methods around minima of the loss landscape to gain insight into which minima are attractors for the dynamics. Crucially, we focus on a notion of data coherence [Dexter et al., 2024] that captures how similar or aligned the contributions of different training examples are to the local curvature of the loss. This measure serves as a bridge between data geometry and the stability of minima: intuitively, solutions where many examples share common "directions" in parameter space (high coherence) are more stable under SGD dynamics, whereas solutions that fit each example independently (low coherence) are less stable. In turn, we prove that the emergence of an implicit simplicity bias that is introduced based on which minima are stable vs unstable and leads SGD to favor simpler solutions that utilize shared features across data points instead of memorizing idiosyncrasies of individual examples.

Furthermore, our framework allows us to compare standard SGD with two variations: a simple *random perturbation* method (which injects isotropic noise during training) and SAM. We find that injecting small random perturbations has essentially the same stability criteria as plain SGD, indicating that when viewed through the lens of linear stability, it does not fundamentally change which minima are favored but increases the *speed* of escape from unstable minima. In contrast, SAM imposes a stricter stability requirement: it penalizes directions with high curvature via an effective Hessian factor in the update. As a result, our analysis predicts that SAM will actively avoid sharper (narrow) minima even when SGD might be marginally stable there, and instead SAM will gravitate even more strongly toward not just flatter, but more highly coherent solutions. In effect, SAM amplifies the simplicity bias inherent in SGD by further discouraging solutions that depend on complex, fragile combinations of features. Notably, this extends the conventional view of SAM as merely favoring flat minima: our analysis shows that it also induces a bias toward solutions of lower complexity that rely on shared structure across examples.

**Contributions.** Our work helps paint a coherent picture in which the "flat minima $\Rightarrow$ good generalization" heuristic and the "SGD finds simple functions" heuristic are two sides of the same coin. We articulate this connection rigorously and in doing so, also suggest that interventions like SAM, which further insist on flatness, are effectively heightening the simplicity bias – an interpretation consistent with recent findings that SAM-trained models prefer more parsimonious representations [Andriushchenko et al., 2023], can enlarge the regime of benign overfitting when compared to SGD [Chen et al., 2023] and can empirically choose better generalizing minima even there are multiple minima with the same flatness [Wen et al., 2023a]. Our main contributions can be summarized as follows:

- **Unified Linear Stability Analysis.** By linearizing the training dynamics around a candidate minimum, we derive the first known precise stability conditions for a randomly perturbed variant of SGD, and SAM to explain when a solution will be an attractor under each algorithm. We examine a data-dependent coherence matrix that measures the alignment between per-example loss Hessians. We prove that the spectral properties of this coherence matrix directly govern stability: roughly, solutions where training examples yield highly aligned curvature (high coherence) remain stable under larger learning rates.

- **Matching lower bounds.** To show tightness of our analysis, we derive matching lower bounds on the stability trace under SAM, further cementing exponential divergence when coherence and curvature align unfavorably.

- **Emergence of Simplicity Bias for SGD.** We prove that if the training data admits a "simple" global solution (where the model uses a few common set of features for many examples), the solution will exhibit high coherence and thereby strong stability, causing SGD to prefer it over more complex solutions (which have lower coherence and are unstable under similar conditions). This result bridges the gap between data geometry and the classical flatness-generalization argument, as highly coherent solutions tend to be flatter in the aggregate sense.

- **SAM intensifies the Simplicity Bias.** Our analysis indicates that SAM's update rule effectively makes it even harder for solutions with disparate, high-curvature directions to remain stable. Consequently, we show that the data coherence explains why SAM not only

finds flatter minima than SGD, but also drives the model toward using more aligned (and hence fewer) features. This aligns with recent empirical observations that SAM leads to simpler or more generalizable representations in deep models [Andriushchenko et al., 2023, Wen et al., 2023a, Springer et al., 2024].

- **Empirical Validation.** We validate our theoretical insights on a two-layer ReLU network, where we can analytically characterize different types of solutions. We prove, for instance, that in a two-layer network, a solution that memorizes each training example in a separate neuron corresponds to a diagonal coherence matrix (no shared features), whereas a solution that generalizes by using common features yields off-diagonal coherence and a dominant principal component. Our results confirm that SGD (and especially SAM) is unlikely to converge to the memorizing solution when a simpler one exists, consistent with the simplicity bias phenomena observed in practice.

## 2  Background

We begin by introducing the linear stability framework, which forms the foundation for our analysis. Linear stability provides a principled way to analyze the local behavior of iterative optimization algorithms in the vicinity of critical points (e.g., local minima or saddle points) by linearizing the update dynamics. This framework has been used to study convergence properties of SGD and its variants and has recently emerged as a useful tool to characterize generalization-relevant behavior such as the ability to escape sharp minima [Wu et al., 2018, Dexter et al., 2024].

**Linearized Dynamics near a Minimum.** Consider a twice-differentiable loss function $L(w)$ over model parameters $w \in \mathbb{R}^d$, and suppose $w^\star$ is a local minimum. Let $\delta_t = w_t - w^\star$ be the perturbation from the minimum at time $t$. Expanding the gradient in a Taylor series around $w^\star$, we obtain

$$\nabla L(w_t) = \nabla L(w^\star + \delta_t) \approx \nabla^2 L(w^\star)\delta_t,$$

since $\nabla L(w^\star) = 0$. For a generic optimization algorithm with update rule $w_{t+1} = w_t - \eta g_t$, the linearized dynamics become

$$\delta_{t+1} = (I - \eta H_t)\delta_t,$$

where $H_t$ is an approximation to the local curvature (e.g., the Hessian or a stochastic surrogate). In the case of stochastic gradient descent (SGD), the curvature matrix $H_t$ is typically estimated using a mini-batch of training samples. Let $\mathcal{S}_t$ denote a randomly sampled batch of size $B$ and define the stochastic Hessian estimate as

$$H_t = \frac{1}{B} \sum_{i \in \mathcal{S}_t} H_i, \quad \text{where } H_i = \nabla^2 \ell(w; x_i, y_i).$$

Then, the SGD update becomes

$$w_{t+1} = w_t - \eta H_t w_t = (I - \eta H_t)w_t = \hat{J}_t w_t, \tag{1}$$

where we use $\hat{J}_t$ to denote the random linear operator at iteration $t$. For deterministic full-batch gradient descent, we replace $H_t$ with the full Hessian $H = \frac{1}{n} \sum_{i=1}^n H_i$ and drop the hat notation.

To study the long-term behavior of the iterates, we analyze the expected squared norm of the weights:

$$\mathbb{E}[\|w_k\|^2] = \mathbb{E}[w_0^\top \hat{J}_1^\top \cdots \hat{J}_k^\top \hat{J}_k \cdots \hat{J}_1 w_0] = \mathbb{E}[\operatorname{Tr}(\hat{J}_k \cdots \hat{J}_1 w_0 w_0^\top \hat{J}_1^\top \cdots \hat{J}_k^\top)].$$

Assuming $w_0 \sim \mathcal{N}(0, I)$, we reduce to analyzing the quantity $\mathbb{E}[\operatorname{Tr}(\hat{J}_k^\top \cdots \hat{J}_1^\top \hat{J}_1 \cdots \hat{J}_k)]$, which captures the contraction or expansion behavior of the iterates under the sequence of update matrices. See more details discussion of assumption in appendix A.

**Stability criterion.** The system is said to be *linearly stable* at $w^\star$ under a given optimization method if the expected squared norm $\mathbb{E}[\|w_k\|^2]$ remains bounded as $k \to \infty$. A sufficient condition for this is that the spectral norm of the average update matrix $\mathbb{E}[\hat{J}_t^\top \hat{J}_t]$ is strictly less than 1. For full-batch gradient descent, this reduces to requiring $\eta < 2/\lambda_{\max}(H)$.

More generally, in the presence of stochasticity and structure in the data, one can derive stability conditions involving both the Hessian spectrum and how curvature is distributed across examples. This motivates the use of a data-dependent coherence measure, which we introduce next.

**Definition 1.** *Coherence measure [Dexter et al., 2024]. For a collection of per-example Hessians* $\{H_i\}_{i=1}^n$, *define the coherence matrix* $S \in \mathbb{R}^{n \times n}$ *with entries* $S_{ij} = \|H_i^{1/2} H_j^{1/2}\|_F = \sqrt{\mathrm{Tr}(H_i H_j)}$. *The coherence measure* $\sigma$ *is defined as follows:*

$$\sigma = \frac{\lambda_{\max}(S)}{\max_{i \in n} \lambda_{\max}(H_i)} \tag{2}$$

High coherence corresponds to strong alignment in the curvature directions induced by different training examples. Intuitively, perturbations in shared directions lead to large changes in loss across many samples, which creates a stronger restorative gradient and thus stabilizes the solution. This measure plays a central role in our analysis: we show that both SGD and SAM favor solutions with high coherence, and that SAM in particular exhibits stronger divergence from low-coherence solutions due to its amplified curvature penalty. The next section builds on this foundation to characterize the dynamics of SGD, random perturbation, and SAM using linear stability theory.

## 3 Main Results

### 3.1 SGD under Random Perturbation

To better understand how optimization algorithms behave near critical points, we begin by analyzing a simple but illustrative baseline: random perturbation-based SGD [Bisla et al., 2022]. This variant injects additive noise into each update step and has been used to improve generalization or to escape sharp minima:

$$\begin{aligned} w_{t+1} &= w_t - \eta \nabla_S l(w + \delta_t) \\ &= w_t - \eta H_t w_t - \eta H_t \delta_t \end{aligned} \tag{3}$$

While this method has been empirically studied, its behavior under the linear stability framework has not been formally characterized. Our goal is to quantify how the injected noise affects both the convergence region and the escape rate from unstable minima. By addressing these questions, we establish a baseline against which we can evaluate SAM's behavior. The following theorem provides bounds on the trajectory norm under random perturbations, showing how noise modifies the divergence behavior without altering the stability threshold.

**Theorem 3.1.** *Given update rule* (3),

1. *Sufficient condition for divergence is as follows:*

$$\eta \geq \frac{\sigma}{\lambda_1} \left(\frac{n}{b} - 1\right)^{-\frac{1}{2}}$$

2. *(Comparative Divergence Speed) Suppose* $\mathrm{Tr}[J^{2k}] \leq C_0 \alpha^k$ *for some constants* $C_0$ *and* $\alpha_k$, *then the divergence rate of the random perturbation method is asymptotically within a constant factor of that of standard SGD:*

$$\lim_{k \to \infty} \frac{E[\|w_k\|^2]_{Random, \, lower \, bound}}{E[\|w_k\|^2]_{SGD, \, lower \, bound}} = \mathcal{O}(1)$$

3. *Suppose the step size satisfies the convergence criterion established in prior stability analyses (e.g., Dexter et al. [2024]). Then, under the random perturbation update* (3), *the expected squared norm of the iterates remains bounded as* $k \to \infty$:

$$\lim_{k \to \infty} E[w_k^T w_k]_{upper \, bound} = \mathcal{O}(1)$$

**Discussion.** Theorem 3.1 characterizes the behavior of random perturbation under the linear stability framework. We find that the divergence threshold for instability remains unchanged from standard SGD as derived by Dexter et al. [2024] (part 1), implying that adding random noise does not alter which minima are stable. However, once a minimum is unstable, the injected noise causes the iterates to diverge at a constant faster rate (part 2). This observation aligns with the intuitive role of noise in facilitating exploration during training. Further, in the stable regime, the iterates do not

converge exactly to the minimum, but instead remain in a bounded region around it (part 3). This residual variance arises from the persistent noise and illustrates a tradeoff: random perturbation aids exploration but limits precision. Together, these results position random perturbation as a useful baseline control for SAM as we now show that SAM's behavior is due to a fundamentally different mechanism that biases optimization toward aligned, lower-complexity minima.

## 3.2 Sharpness aware minimization (SAM)

We now analyzing SAM, an algorithm explicitly designed to seek flatter minima by optimizing a worst-case perturbed loss. SAM replaces the standard gradient descent step with a descent direction that maximizes the loss within a neighborhood of the current iterate. Formally, the gradient can be approximated as:

$$\nabla l(w)_{\text{SAM}} \simeq \nabla l(w + \rho \frac{\nabla l(w)}{||\nabla l(w)||})$$

Under the quadratic setting, we can write the iterate update process as:

$$\begin{aligned}
w_{t+1} &= w_t - \eta \nabla_S l(w_t + \rho \frac{\nabla l(w_t)}{||\nabla l(w_t)||}) \\
&= (I - \eta H_t(I + \frac{\rho}{||Hw_t||}H))w_t
\end{aligned} \tag{4}$$

While the gradient update in SAM admits a closed-form expression under the quadratic approximation, the presence of the norm in the denominator—i.e., $||Hw_t||$—introduces significant analytical challenges due to its dependence on the current iterate. To facilitate tractable analysis, we follow prior works [Andriushchenko and Flammarion, 2022, Du et al., 2022, Zhou et al., 2025] and replace this quantity by a fixed scalar value $\alpha$. This simplification allows us to isolate the effect of the curvature term and focus on the directional dynamics introduced by the sharpness-aware perturbation. Under this update, the SAM update reduces to a linear transformation governed by an effective Hessian of the form $H_{\text{SAM}} = H\left(I + \frac{\rho}{\alpha}H\right)$ which portrays a *stricter stability criterion* and fundamentally different optimization dynamics than SGD. This allows SAM to not only escape sharp minima more aggressively but also to selectively stabilize solutions that exhibit *coherent curvature*, thereby amplifying simplicity bias. In our theory, we analyze the noise arising from the alignment of the space spanned by different sample which accumulate over steps. For the following, we provide simplified version of our theory (see Appendix C.8 and Appendix C.9 for exact details).

**Theorem 3.2** ((Simplified) Linear Stability of SAM). *Consider the update rule of SAM under a quadratic loss approximation:*

$$w_{t+1} = \left(I - \eta H_t\left(I + \frac{\rho}{\alpha}H\right)\right) w_t,$$

*where $H_t$ is the mini-batch Hessian at time $t$, $\rho$ is the SAM perturbation radius, and $\alpha$ is a fixed approximation to $||Hw_t||$.*

1. ***Divergence criterion.*** *SAM diverges if the largest eigenvalue of the Hessian exceeds the following threshold:*

   $$\lambda_{\max}(H) \geq \frac{\sigma}{\eta}\left(\frac{n}{B} - 1\right)^{-1/2}\left(1 + \frac{\rho}{\alpha}\lambda_{\min}(H)\right)^{-1}.$$

   *Compared to SGD, this condition is stricter due to the additional curvature-dependent term in the denominator, implying that SAM escapes sharp minima more aggressively.*

2. ***Convergence criterion.*** *If there exists $\epsilon \in (0,1)$ such that*

   $$\frac{\epsilon}{\eta} \leq \lambda_i + \frac{\rho}{\alpha}\lambda_i^2 \leq \frac{2-\epsilon}{\eta}, \quad \forall i \in [d],$$

   *and the accumulated noise decays sufficiently fast (see Appendix C.9), then the iterates converge in expectation:*

   $$\lim_{k \to \infty} \mathbb{E}[||w_k||^2] = 0.$$

see the divergence part proof in Appendix C.8 and see the convergence part of proof in Appendix C.9

**Discussion:** Theorem 3.2 shows that the optimization dynamics of SAM are also governed by the coherence measure that captures how sample gradients align. However, if we map the SAM dyanamics to those of SGD as per Eq (1), we can extrapolate that SAM operates on a sharper surface where the effective Hessian and coherence matrix becomes

$$H_{\text{SAM}} = \left(I + \frac{\rho}{\alpha}H\right)H,$$

$$S_{\text{SAM},ij} = \sqrt{\text{Tr}[(I + \frac{\rho}{\alpha}H)H_i(I + \frac{\rho}{\alpha}H)H_j]}.$$

SAM's update effectively introduces an additional Hessian-dependent factor – penalizing directions with large curvature – which tightens the stability criterion. Qualitatively, a solution that might be marginally stable under SGD (with eigenvalues just below the SGD stability threshold) can become unstable under SAM if even a single eigen-direction has curvature beyond SAM's narrower tolerance. This theoretical insight aligns well with SAM's design goal of seeking flat minima [Foret et al., 2021], but it goes further by pinpointing which sharp minima are especially disfavored: namely, those where the sharpness arises from directions that are not supported uniformly by all training examples. By contrast, if a sharp curvature direction is "universal" across examples (high coherence), it is somewhat mitigated in our stability condition – intuitively, SAM is less alarmed by curvature that stems from a feature direction that all data agree on, than by curvature that comes from one-off fluctuations. In practical terms, this means SAM biases training even more strongly toward solutions that rely on global features of the data. Solutions that depend on any single example (or a small subset of examples) in a unique way will tend to have some high-curvature direction localized to that example's loss, making them unstable under SAM even if SGD might have tolerated them. Further, SAM also escapes exponentially faster from minima that it deems as unstable.

We now show the optimality of the SAM divergence bound with a matching lower bound:

**Theorem 3.3.** *For every choice of* $\lambda_1 > 0, n \in N, B \in [n], \eta > 0$ *and* $\sigma \in [n]$*, that satisfied*

$$\lambda_1(1 + \frac{\rho}{\alpha}\lambda_1) \leq \frac{2\sigma}{\eta}(\sigma + \frac{n}{B} - 1)^{-1}, \tag{5}$$

*there exist a set of PSD matrices* $\{H_i\}_{i \in [n]}$ *such that* $\lambda_{\max}(H) = \lambda_1$ *and* $\lim_{k \to \infty} E||\hat{J}_k...\hat{J}_1||_F < n$

## 3.3 Emergence of Simplicity Bias: Realization in Two-Layer ReLU Models

We now instantiate the linear stability framework in a concrete neural network setting to analyze how the theoretical insights developed so far translate to realistic model architectures. Specifically, we focus on a two-layer ReLU network trained with mean squared error (MSE) loss:

$$f_w(x) = W_2 \cdot \text{ReLU}(W_1 x + b),$$

where $W_1 \in \mathbb{R}^{d_2 \times d}$ maps inputs to hidden units, $W_2 \in \mathbb{R}^{1 \times d_2}$ maps activations to output, and $b \in \mathbb{R}^{d_2}$ is a bias term.

This setting allows us to bridge abstract notions such as curvature, coherence, and stability with concrete properties of network solutions — including memorization, feature sharing, and low-rank structure. We adopt a synthetic data distribution from prior work [Wen et al., 2023a] where inputs $x \in \{-1, 1\}^d$ are drawn i.i.d. and labels are generated $y = x[0]x[1]$, ensuring that both simple and complex solutions exist.

Crucially, we analyze the Hessian of the MSE loss under exact interpolation. In this regime, the gradient of the loss at the optimum is zero and the Hessian simplifies to:

$$H = \frac{1}{n}\sum_{i=1}^{n} \nabla f_w(x_i)\nabla f_w(x_i)^{\top},$$

enabling direct application of our earlier stability results. By examining how per-sample gradients align across examples, we quantify the coherence structure of different solutions and show how this governs their stability under SGD and SAM.

**Memorization and Coherence.**   We begin by characterizing memorization in our setup – a data point is *memorized* if it activates a unique hidden neuron [Wen et al., 2023a]. A *memorized solution* is one where each training point is memorized. A *generalizing solution* reuses the same features (neurons) across multiple samples, resulting in aligned gradients and higher coherence. The following result formalizes the relationship between memorization and coherence.

**Theorem 3.4** (Coherence Characterization of Memorization). *A two-layer ReLU network defines a memorizing solution if and only if the coherence matrix $S$ is diagonal.*

This observation implies that in the memorizing regime, the sample-wise Hessians are orthogonal, i.e., $\text{Tr}[H_i H_j] = 0$ for $i \neq j$. As a result, the coherence matrix reduces to a diagonal form, eliminating any cross-sample interaction. Consequently, the corresponding stability condition degrades to its weakest form, as no spectral amplification arises from off-diagonal terms.

In contrast, generalizing solutions induce non-trivial off-diagonal components in $S$, reflecting shared activation patterns or feature overlap across examples. This coherent structure enhances the dominant eigenvalue of $S$, thereby tightening convergence guarantees and improving stability under both SGD and SAM. Hence, any statistical or geometric correlation among training samples inherently rules out the memorizing case and promotes more favorable optimization dynamics.

**Constructing Generalizing Solutions.**   To systematically explore generalization within this framework, we define a family of $(C, r)$-generalizing solutions, where $C$ controls the number of active features and $r$ determines the sharpness of the solution. For fixed $r$, all such solutions are equally flat in trace norm, but differ in complexity through $C$.

**Definition 2.** *($(C, r)$-generalizing solution) Let $\{a_1, a_2, \ldots, a_C\} \in \{0, 1\}^C$. We construct $W_1$ such that each hidden unit encodes a pattern of the form:*

$$W_{1,j} = r \cdot [(-1)^{a_1}, (-1)^{a_2}, \ldots, (-1)^{a_C}, 0, \ldots, 0],$$

*with $j$ indexing a binary encoding of the $a_i$'s i.e. $j = 1 + \sum 2^{i-1} a_i$. We set $W_2[j] = \frac{1}{r}(-1)^{a_1 + a_2}$ to match. For $k > C$, $W_{1,k} = 0$, $W_2[k] = 0$, $b[k] = 0$.*

It is straightforward to verify that $\text{Tr}[H_i] = \frac{1}{r^2}(d+1) + r^2$, indicating that $r$ controls the flatness of the solution. The minimum trace—and thus the flattest solution—is achieved when $r = (d+1)^{1/4}$. For any fixed $r$, the overall flatness $\text{Tr}(H)$ remains constant regardless of $C$. The weights $W_1$ exhaustively encode all possible feature values, and the bias $b$ ensures that $\text{ReLU}(W_1 x + b)$ activates only one row that match $x$. Given that $y = x[0]x[1]$ (as described in Section 3.3), the construction guarantees that each $(C, r)$-generalizing solution is also interpolating: $f(x_i) = y_i$ for all $i \in [n]$. In this framework, $C$ acts as a complexity surrogate—controlling the number of features used—while preserving identical flatness across interpolating solutions. An illustrative example is provided in Appendix C.1. As we show next, the coherence matrix spectrum depends heavily on $C$:

**Theorem 3.5** (SGD Stability of $(C, r)$-Generalizing Solutions). *Fix $r = (d+1)^{1/4}$. Then, with probability at least $1 - \delta$, for a randomly drawn dataset of size $n$, the top eigenvalue of the coherence matrix under a $(C, r)$-generalizing solution satisfies:*

$$\lambda_{\max}(S) = \mathcal{O}\left(\frac{n}{2^C}(d+1)^{1/2}\right),$$

*while $\max_i \lambda_{\max}(H_i) = 2(d+1)^{\frac{1}{2}}$.*

(See Appendix C.11 for dependency of $\delta$) combining with Theorem for the SGD bounds [Dexter et al., 2024], this implies that simpler (low-$C$) generalizing solutions are more stable under SGD, even when all candidate solutions lie in equally flat regions of the loss.

**SAM Favors Simpler Solutions.**   Finally, we analyze how SAM further shifts this preference. Under SAM, the coherence matrix becomes

$$S_{ij}^{\text{SAM}} = \sqrt{\text{Tr}\left[(I + \frac{\rho}{\alpha}H)H_i(I + \frac{\rho}{\alpha}H)H_j\right]},$$

which accentuates the interaction between aligned directions. As a result, the top eigenvalue of $S^{\text{SAM}}$ grows more sharply for generalizing solutions with small $C$ than for more complex ones.

This bias is formalized in the following bound:

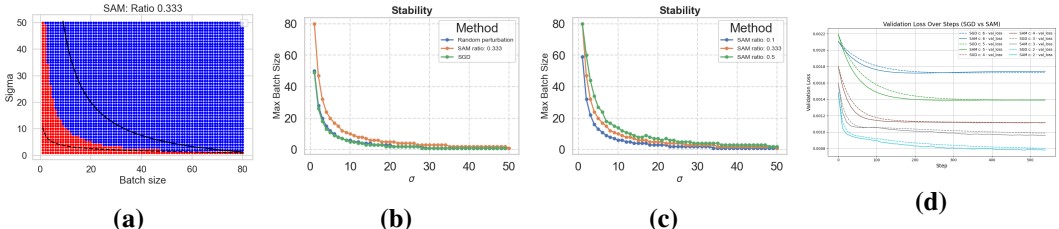

Figure 1: **Comparison of optimization dynamics across different methods and configurations.**
(a) SAM's dyanamics over different hyper-parameter settings (Red:diverging Blue:converging). (b) **Boundary comparison: SGD, random perturbation, SAM.** SGD and random perturbation boundaries largely overlap, while SAM diverges in more combination of batch size and $\sigma$. (c) **SAM boundaries at different $\frac{\rho}{\alpha}$:** Higher $\frac{\rho}{\alpha}$ further tightens the SAM boundary between the converging and diverging regimes. (d) **Convergence under different $C$, fixed $r$:** For 2-layer ReLU networks, SAM converges faster across varying $C$ with fixed $r$.

| Method | Coherence Measure | $\lambda_{\max}(S)$ | ER | $\max_i \lambda_{\max}(H_i)$ | $\lambda_{\max}(H)$ | $\mathrm{Tr}(H)$ |
|---|---|---|---|---|---|---|
| SGD | 133.942 | 12740.285 | 6.167 | 94.385 | 6.776 | 51.954 |
| SAM $\rho = 0.01$ | 121.473 | 10103.288 | 6.2 | 82.882 | 6.211 | 48.401 |
| SAM $\rho = 0.05$ | 90.309 | 6421.689 | 6 | 70.948 | 4.988 | 38.481 |
| SAM $\rho = 0.1$ | 65.656 | 3445.964 | 5.6 | 52.294 | 3.834 | 29.140 |

Table 1: Coherence and Hessian-based metrics across methods. **ER** stand for effective rank of the features. We perform PCA on the features of training data and set the threshold for effective rank to be 90 percent to determine the activation pattern used in different combination of parameters and optimization.

**Theorem 3.6** (SAM Stability of $(C, r)$-Generalizing Solutions). *Under a $(C, r)$-generalizing solution for a randomly iid drawn dataset of size n, the top eigenvalue of the SAM-induced coherence matrix satisfies:*

$$\lambda_{\max}(S^{\mathrm{SAM}}) = \mathcal{O}\left( \frac{n}{2^c}(d+1)^{\frac{1}{2}} \sqrt{(1 + \frac{\rho}{\alpha}\frac{2(d+1)^{\frac{1}{2}}}{2^c})^2 + \frac{\rho^2}{\alpha^2}(\frac{1}{n}(\frac{1}{2^c} - \frac{1}{2^{2c}}))4(d+1)} \right),$$

*and,*

$$\max_i \lambda_{\max}((I + \frac{\rho}{\alpha}H)H_i) = 2(d+1)^{\frac{1}{2}}(1 + \frac{\rho}{\alpha}\frac{1}{2^C}2(d+1)^{\frac{1}{2}})$$

Combining with Theorem 3.2, Theorem 3.6 shows that SAM not only escapes sharp minima more aggressively, but also *amplifies stability differences between simple and complex solutions* by favoring solutions with smaller $C$ more aggressively, further biasing optimization toward lower-complexity minima. This explains recent empirical observations of SAM inducing low-rank or structured representations even when curvature alone does not distinguish between candidate solutions.

## 4  Experiments

In the experimental section, we empirically validate the following key aspects of our theoretical framework: (1) The behavior of SAM, random perturbation, and SGD under varying combinations of batch size $B$ and noise scale $\sigma$ to validate our theoretical findings. (2) The dynamics of different optimization algorithms in the vicinity of various $(C, r)$-generalizing solutions, again for validating our theory, (3) The influence of the coherence measure throughout the training process, its role in optimization, and its sensitivity to different training hyperparameters across methods.

**Local: Linear stability in quadratic loss for different algorithms** $(B, \sigma)$ We investigate the diverging/converging behavior compared to our theoretical upper and lower bounds over different values of $(B, \sigma)$ for quadratic loss. For space, the experimental setup is moved to Appendix C.3. The results are presented in Fig 1(a). The dashed line and solid line are plotted as per Theorem 3.3 and Theorem 3.2 respectively. We observe a gap in small batch size setting also apparent from our theory but for larger batch sizes, our theory can accurately predict the diverging and converging

behaviors. For Figures 1(b)(c), we compare the experimental boundary of SGD, random perturb and SAM and observe that the boundary between stable and unstable regimes shift as we increase the ratio $\frac{\rho}{\alpha}$ as also predicted by our theory. Additionally, SGD and random perturbation algorithm show strong overlap in the boundary which also aligns with our theoretical results.

**Local: Linear stability in MSE loss for different algorithms in 2-layer ReLU network.** To study the local behaviors of different algorithms, we initialize the model around the $(C, r)$ solution (Def. 2) with small Gaussian noise $N(0, 0.01)$ to the model weights to ensure non-zero gradients. We sample $n = 100$ data points as described in set up in section 3.3. From figure 1(d), we observe for smaller $C$, the converging speed is indeed faster compared to those with high $C$ as expected from our theory. Additionally, we find that when compared between SGD and SAM, SAM demonstrate higher converging speed which also aligns with our analysis. (See Appendix C.4 for additional details)

**Global: The role of coherence in the training.** In this section, we investigate the role of the coherence measure through the training dynamics using the setup in the previous section. The final results is presented in Table 1. A key finding is that SAM substantially reduces the effective coherence measure compared to SGD. To better understand the representational changes induced by SAM, we also compute the effective rank of the learned features. Specifically, we perform principal component analysis (PCA) on the training features and define the effective rank as the minimum number of components required to explain 90% of the variance. We observe a consistent decrease in effective rank alongside coherence, suggesting that SAM encourages more compact and structured representations – an observation that aligns with our theoretical predictions.

Furthermore, we track the evolution of the coherence measure throughout the training (Figure. 2) and find that it varies dynamically, rather than remaining fixed. This suggests that coherence is a nonstationary quantity during optimization, and tracking its trajectory may yield new insights into the evolving relationship between data samples.

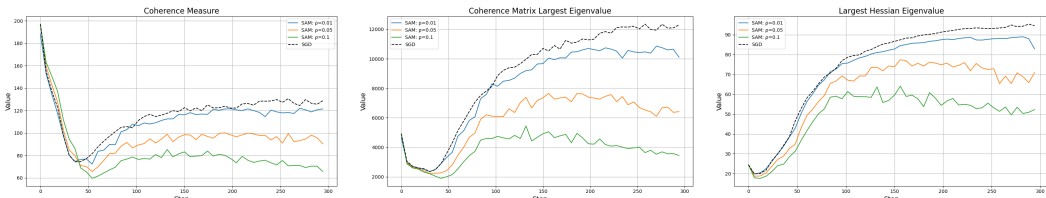

Figure 2: **2-layer ReLU network.** SAM imposes strong regularization on the maximum elementwise Hessian eigenvalue, and this also reduces the largest eigenvalue of the coherence matrix, which implies the stability condition is satisfied with smaller $\sigma$.

**Conclusion.** Our analysis reveals that the stability properties of optimization algorithms—especially in the presence of data coherence—are central to the emergence of generalization and simplicity in deep learning. We show that SAM amplifies this implicit bias by selectively stabilizing flatter, more coherent solutions, offering a theoretical explanation for its empirical success. These results suggest a unifying lens to interpret generalization as a stability-driven selection of solutions, and open avenues for designing optimizers that align algorithmic bias with data geometry.

**Limitation.** A primary limitation of our work is the reliance on a linear approximation of the loss landscape, as our analysis focuses on local behavior near minima. While we empirically explore the relationship between the coherence measure and training hyperparameters using two-layer ReLU networks, extending this investigation to larger models and real-world datasets remains an important direction for future work. Such studies could offer deeper insights into the practical significance of coherence in larger deep learning systems. For more discussion regarding limitation and practical or potential direction, we provide detailed exposition in appendix A.

## Acknowledgments

We thank the Central Indiana Corporate Partnership AnalytiXIN Initiative for their support.

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

# A More discussion on limitation and practical implication

**Assumption and simplification:** Our theoretical analysis is limited to a linear approximation near local minima, which may not capture the full dynamics of training. While our analysis is local, this regime remains highly relevant: empirical studies show that loss landscapes are often locally approximately quadratic, even in overparameterized models [Li et al., 2018], motivating many theoretical frameworks for optimization and generalization [Achour et al., 2024, Wen et al., 2023b]. Since training often proceeds through plateau phases near minima, local dynamics are central to understanding stability. Like gradient flow and neural tangent kernel analyses, our work adopts simplifying assumptions that, while not universally valid, yield valuable insights. Our approach aligns with prior work demonstrating the utility of local approximations.

Other than linear assumption, we also utilize assumption and simplification such as i.i.d. data, two-layer ReLU networks, and restrict ourselves to basic optimizers (SGD, SAM). We offer clearification for the above assumption in following:

First, the i.i.d. sampling assumption is standard in theoretical deep learning (e.g., generalization bounds, stochastic processes) and reflects how minibatches are typically drawn in practice. Studying non-i.i.d. settings like curriculum or imbalanced sampling is a valuable direction for future work.

Second, while (c,r)-generalizing solutions are restricted, they reflect key structures observed in deep networks—especially in activation patterns. ReLU networks exhibit rich symmetries (e.g., permutation, rescaling) that yield exponentially many equivalent solutions with identical loss, gradient, and Hessian, making our framework practically relevant despite its constraints. Such simplifications are common to enable tractable analysis.

Third, we focus on plain SGD and SAM to isolate the roles of noise and sharpness in a clean setting. Extending to optimizers like momentum SGD or Adam, which interact with curvature in subtle ways, is a promising direction that can reveal more optimization properties under data geometry structure.

Lastly, our analysis reveals a limitation of optimization dynamics in settings where the data exhibits high redundancy or strongly correlated features. In such cases, the linear stability criteria become less sensitive to solution complexity, as many parameter configurations yield similar activation patterns and induce comparable coherence measures. This results in a potential blind spot: the optimization algorithm may fail to distinguish between simple and complex solutions if the underlying data geometry does not sufficiently break symmetry. Recognizing this limitation offers a valuable direction for future theoretical work, particularly in understanding how optimization behavior is shaped not only by the loss landscape but also by the structure and diversity of the training data.

**Practical Implications:** In terms of practicality of the proposed coherence measure, it is computationally expensive and not typically used in training. Our work is intended as a first step in highlighting coherence as a potentially valuable conceptual lens for understanding training dynamics and generalization. While the coherence measure is not yet a standard diagnostic tool, this is also true for many theoretical quantities when first introduced. A relevant example is sharpness (e.g., maximum eigenvalue of the Hessian), which was originally difficult to compute at scale, yet it inspired successful optimization strategies such as SAM that approximate the principle indirectly. Similarly, although computing exact gradient coherence or per-sample Hessian quantities is currently expensive, we see this as a motivation for future work on scalable surrogates or proxies. Our contribution is to show that coherence connects to stability in a theoretically grounded way, suggesting that it could inform the design of future optimizers or monitoring tools, even if not computed directly.

For potential direction for design of algorithm, one concrete idea is to adapt the learning rate based on coherence between mini-batches—reducing it when gradients are aligned, and increasing it when they diverge. Other possibilities include regulator that promote gradient alignment or batch selection strategies favoring coherence. While we have not explored these experimentally, we view them as promising directions for future work.

**Larger Empirical Scope:** To investigate the scalability and practical relevance of our insights, we conducted additional experiments on the CIFAR-10 dataset using a ResNet-18 model (11.7M parameters). To approximate the coherence measure in a tractable way, we used the pairwise dot product of per-sample gradients normalized by the maximum gradient norm: $\frac{\nabla l_i \nabla l_j}{\max \|\nabla l_k\|}$. For further approximation, We compute this on a fixed subset of 100 samples (10 per class) to construct a coherence matrix. For the feature rank calculation, we record the features from before the last linear

layer for whole training dataset and use PCA to calculate the rank of feature with explain ratio up to 99.9 percent. According to our experiments, we find that the feature rank decrease as expected and the the approximated coherence measure also closely follow the training process despite milder trends due to the approximation. The results are shown in following:

| Optimizer | Rank | Coherence |
|---|---|---|
| SGD | $148.33 \pm 3.22$ | $1.0045 \pm 0.0020$ |
| SAM (0.05) | $157.67 \pm 9.29$ | $1.0052 \pm 0.0060$ |
| SAM (0.1) | $144.33 \pm 7.10$ | $1.0771 \pm 0.0680$ |
| SAM (0.2) | $128.67 \pm 5.51$ | $1.0907 \pm 0.0890$ |

Table 2: Effect of Optimizer and SAM radius on feature rank and coherence. We record the rank and coherence on subset of CIFAR10 after training for 200 epochs.

# B  Related Work

**Flat vs. sharp minima and generalization.** The connection between the geometry of minima and generalization in deep networks has been studied extensively. Hochreiter and Schmidhuber [1997] first argued that flat minima (regions in parameter space where the loss remains low) correspond to better generalization, while sharp minima might lead to worse generalization. Keskar et al. [2017] provided empirical evidence that large-batch SGD tends to find sharper minima than small-batch SGD, correlating with higher test error, bringing this idea to prominence. However, Dinh et al. [2017] pointed out that the sharpness of a minimum is not an invariant property (reparameterizations of the model can change the Hessian spectrum without affecting generalization), cautioning that one must carefully define "sharpness" (e.g., by normalizing for scale or using local subspace measures). Our work incorporates this perspective by focusing on a *relative* stability analysis: effectively, we look at sharpness in the context of the optimizer's step size and algorithm, which is invariant to certain rescaling (for example, SAM's notion of sharpness implicitly accounts for parameter scale through the perturbation magnitude).

Sharpness-Aware Minimization [Foret et al., 2021] and follow-up methods (e.g., adaptive SAM by Kwon et al., 2021, investigations in Chen et al., 2023) directly encode flatness into the training objective. By explicitly favoring flatter minima, SAM biases the training trajectory toward solutions that are less sensitive to perturbations in parameter space [Zhang et al., 2024, Chen et al., 2023]. Empirically, SAM has demonstrated improved generalization across many tasks. The work of Andriushchenko et al. [2023] is particularly relevant to our findings: they show that SAM not only finds flatter minima but that the learned features (e.g., the covariance of layer activations) tend to be lower rank, suggesting the model focuses on a smaller set of principal components of the data. This aligns with our result that SAM bias can lead to simpler (more coherent) feature usage. There have also been studies connecting flatness to other measures like noise stability: Jiang et al. [2020] evaluate a variety of complexity measures (including some Hessian-based) to see which best predict generalization; they found that no single measure works universally, but a combination can. Our introduction of coherence could add a new dimension to such measures, since it incorporates data-dependent interactions.

**Linear stability.** Linear stability has gained increasing attention in recent machine learning research as a tool to characterize the local convergence or divergence behavior around minima. This framework enables a unified perspective that jointly considers the data distribution, loss landscape geometry, and optimization dynamics. Prior works such as Wu et al. [2018, 2022], Wu and Su [2023] leveraged linear stability to analyze how noise interacts with local minima and to derive convergence criteria based on the Frobenius norm of the Hessian and Ma and Ying [2021] use the framework of linear stability to study property of noise in terms of it higher order moment. More recently, Dexter et al. [2024] introduced a coherence-based measure that captures fine-grained alignment properties of the data through the Hessian. These lines of work provide valuable new perspectives on the interplay between data and optimization—perspectives that are difficult to obtain through classical optimization analysis alone—and offer a deeper understanding of local training dynamics.

Our work is closely related to [Wu et al., 2018, 2022, Dexter et al., 2024, Wu and Su, 2023, Mulayoff and Michaeli, 2024]. Compared to Dexter et al. [2024] who focused on the analysis of SGD, we

take one step further and analyze random noise injected SGD and SAM. Specifically, we investigate how SAM influences local optimization dynamics and how it interacts with the structure of the data. Furthermore, we provide an explicit analysis of two-layer ReLU networks, revealing connections between linear stability, neural activations, and solution stability. This helps elucidate the role of SAM in shaping both the geometry and generalization behavior of trained models. Finally, unlike these prior works, we also set up realization of the theory to a two layer neural network and discuss how the insight from analysis in linear stability can be transferred to the neural network and show how the pattern of activation in neural network can related to result of linear stability. Specifically, we explicitly construct a 2-layer neural network with several solutions of the same sharpness but different complexity (captured by the sparsity in the activation pattern), and show that SGD (and SAM even more aggressively) prefers simpler (sparser) solutions.

**Stability and implicit bias in optimization.** Our use of "stability" is in the sense of dynamical stability of fixed points for the parameter update. This differs from the notion of algorithmic stability in learning theory (e.g., Hardt et al., 2016), which concerns how sensitive the final model is to removal of a training example. Algorithmic stability yields generalization bounds but doesn't directly explain which solution is picked. Nonetheless, both concepts are linked: an optimizer that always returns the same minimum despite small data perturbations might be one that has a strong attractor basin (stable solution).

A large body of work on implicit bias of gradient methods has focused on linear models or homogeneous models, proving that gradient descent converges to particular norm-minimizing solutions or maximum margin solutions [Soudry et al., 2018, Gunasekar et al., 2017]. For example, Soudry et al. [2018] show that for linearly separable data and logistic loss, SGD converges to the max-$L_2$-margin classifier. This can be seen as a form of simplicity bias (since a max-margin separator in linear space is a simpler decision boundary than a complex wiggle that also separates the data). In deep networks, Lyu and Li [2020] extended this to deep homogeneous networks (showing convergence to margin maximization). These works explain *which* solution among the continuum of minimizers is chosen, in terms of margins or norms. Our work provides a complementary lens: rather than characterizing the final solution in closed-form, we explain it via the dynamics preferences (coherence and stability during training). Margin and flatness might be connected; indeed, a large margin classifier often corresponds to a broad basin in loss landscape. Exploring the link between coherence and margin could be interesting (perhaps high coherence solutions also align with large margin in classification tasks).

The notion of *simplicity bias* has been documented empirically by several works. Arpit et al. [2017] found that deep nets first fit the "easy" patterns (e.g., clean labels) before memorizing noisy data, indicating a bias towards simpler functions. Kalimeris et al. [2019] and Valle-Pérez et al. [2019] argued from a information/combinatorics perspective that, because there are exponentially more complex functions than simple ones, a random initialization plus SGD is more likely to land in a simple function that fits the data (if such exists). Shah et al. [2020] (Pitfalls of Simplicity Bias) constructed datasets with multiple features to quantify this bias and showed it can hurt robustness. Our results give a theoretical underpinning to these observations by linking them to the Hessian structure and training dynamics: effectively, the simple patterns correspond to directions in which many data points have aligned gradients, hence those get learned quickly and form a stable basis for the solution, whereas complex patterns do not align and either get learned later or not at all.

Recently, Morwani et al. [2023] provided a rigorous analysis of simplicity bias in one-hidden-layer ReLU networks (in the infinite width, lazy training regime). They defined simplicity in terms of the function depending on a low-dimensional projection of inputs and proved that indeed gradient descent finds such low-dimensional solutions under certain conditions. Their findings dovetail nicely with our coherence interpretation (low-dimensional projection usage implies high alignment among gradients of those inputs). While their analysis is specialized to a particular regime, ours aims to be more generally intuitive and spans beyond the NTK regime by considering the Hessian of the nonlinear model.

Another related concept is *Neural Collapse* [Papyan et al., 2020], which describes that at the final layer of a classifier, the class means and features tend to align in certain simple symmetric patterns. Neural collapse occurs in the late phase of training and indicates a sort of self-organization of features. This might be seen as a high-coherence structure in the last-layer gradients for examples of the same

class. While our work did not directly address neural collapse, the idea that training dynamics lead to aligned and symmetric configurations is broadly consistent.

**Data geometry and gradient alignment.** The role of data distribution in learning dynamics has been explored under terms like *gradient confusion* [Sankararaman et al., 2020] and *gradient alignment*. When gradients of different examples are more aligned, training converges faster and perhaps finds simpler models. Sankararaman et al. [2020] demonstrated that increasing overparameterization can reduce gradient confusion (making gradients more aligned by virtue of more flexible models finding a common direction) up to a point, which speeds up convergence. Chatterjee [2020] studied how examples that are hard or easy influence learning; easy examples likely align well with the gradient direction. Our coherence matrix formalizes one aspect of gradient alignment (at a second-order level, but one could similarly define $G_{ij} = \nabla \ell_i^\top \nabla \ell_j$ for first-order gradients). In fact, one could incorporate first-order coherence in our analysis; we focused on Hessian since it directly ties to stability, but gradient dot products matter for the actual update direction in SGD. A high Hessian coherence usually also implies gradient coherence at $w^*$ if $w^*$ is a zero training error solution (gradients are zero at $w^*$, but consider nearby points or earlier in training). Also, this Gram matrix–like definition of alignment appears in several related formulations. In its simplest form, the gradient Gram matrix $G_{ij} = \nabla \ell_i^T \nabla \ell_j$ has been used to quantify gradient diversity, mutual coherence, or feature alignment in analyses of SGD [Sankararaman et al., 2020]. In neural tangent kernel (NTK) theory or fisher kernel terminology[Khanna et al., 2019], a closely related object appears as the empirical kernel matrix, whose $(i, j)$ entry is given by $\nabla_w f(x_i, w)^\top \nabla_w f(x_j, w)$, which offer interpretation for the relationship among data.

In summary, our work synthesizes ideas from these threads: we put forth coherence as a data-dependent quantifier that influences stability of solutions, thereby linking the optimizer's implicit bias to the geometry of data in parameter space. By doing so, we integrate perspectives from flat minima research, implicit bias theory, and empirical studies of feature learning. We hope this unification will spur further research in understanding and controlling the biases of gradient-based training in deep learning.

# C   Appendix – Experiments and Proofs

## C.1   Illustrative example for $(C, r)$ solution and calculation of the $r$ and trace

Recall our construction for $(C, r)$-generalizing solutions. We design $W_1$ by an exhaustive enumeration of all possible feature constructions of size $C$. In other words, $\forall \{a_1, a_2...a_C\} \in \{0, 1\}^C$, let the $j^{\text{th}}$ row of $W_1$ be $W_{1,j} = r[(-1)^{a_1}, (-1)^{a_2}, ...(-1)^{a_C}, 0, 0, ...0]$, with $j = 1 + \sum 2^{i-1} a_i$. Similarly, let $b[j] = -r(C - 1)$. We set $W_2[j] = \frac{1}{r}(-1)^{a_1 + a_2}$. For $k > C$, $W_{1,k} = 0$, $W_2[k] = 0, b[k] = 0$. The following is the $W_1$ with $d = 5$, $c = 3$, $r = 1$ and hidden layer with 10 neurons.

$$\begin{bmatrix} 1 & 1 & 1 & 0 & 0 \\ -1 & 1 & 1 & 0 & 0 \\ 1 & -1 & 1 & 0 & 0 \\ -1 & -1 & 1 & 0 & 0 \\ 1 & 1 & -1 & 0 & 0 \\ -1 & 1 & -1 & 0 & 0 \\ 1 & -1 & -1 & 0 & 0 \\ -1 & -1 & -1 & 0 & 0 \\ 0 & 0 & 0 & 0 & 0 \\ 0 & 0 & 0 & 0 & 0 \end{bmatrix} \tag{6}$$

The following is the $W_2$ with $d = 5$, $c = 3$, $r = 1$ and hidden layer with 10 neurons.

$$\begin{bmatrix} 1 \\ -1 \\ -1 \\ 1 \\ 1 \\ -1 \\ -1 \\ 1 \\ 0 \\ 0 \end{bmatrix} \tag{7}$$

The following is the $b$ with $d = 5$, $c = 3$, $r = 1$ and hidden layer with 10 neurons.

$$\begin{bmatrix} 2 \\ 2 \\ 2 \\ 2 \\ 2 \\ 2 \\ 2 \\ 2 \\ 0 \\ 0 \end{bmatrix} \tag{8}$$

For the following, we show calculation of relationship between trace of Hessian. We first show the trace of one sample and corresponding flatness. As we known in previous calculation that gradient can be as follows:

$$\nabla f_w(x_i) = \begin{bmatrix} \text{ReLU}(W_{1,1}x_i) \\ ... \\ \text{ReLU}(W_{1,d_2}x_i) \\ W_{2,1}\mathbf{1}[W_{1,1}x_i > 0]x_i \\ ... \\ W_{2,j}\mathbf{1}[W_{1,d_2}x_i > 0]x_i \\ W_{2,j}\mathbf{1}[W_{1,1}x_i > 0] \\ ... \\ W_{2,j}\mathbf{1}[W_{1,d_2}x_i > 0] \end{bmatrix} \tag{9}$$

And the Hessian is $\nabla f_w(x_i)\nabla f_w(x_i)^T$. (for zero loss solution) Then the trace will be $\text{Tr}[\nabla f_w(x_i)\nabla f_w(x_i)^T] = ||\nabla f_w(x_i)||^2$ Now, we have exactly one activation at a time due to the bias ($b$) that impose such restriction. Therefore, the $||\nabla f_w(x_i)||^2 = r^2 + \frac{1}{r^2}d + \frac{1}{r^2} = r^2 + \frac{1}{r^2}(d+1)$

## C.2 Role of coherence measure in dynamics

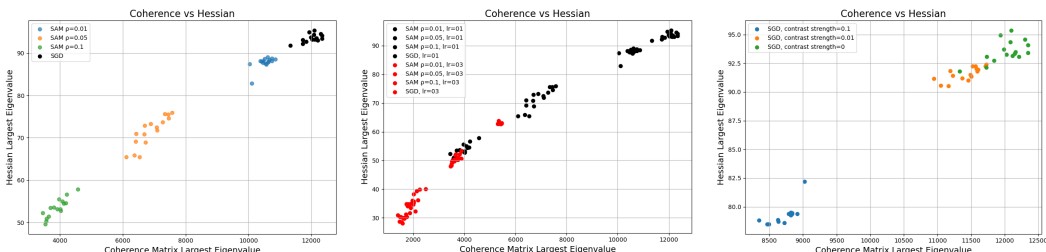

Figure 3: **2-layer ReLU network.** We found that the SAM method can impose strong regulation on the maximum eigenvalue elementwise, and this also reduce the strengthen of the largest eigenvalue of the coherence matrix. It means that the stability condition can be satisfied with smaller $\sigma$. From our experiments, we find that the sharpness of the solution impose strong regulation of the eigenvalue of the coherence matrix.

Figure 4: **2-layer ReLU network.** (**Left**) Comparison of SGD and SAM with different $\rho$. (**Middle**) We perform the same set of experiment with increased learning rate from 0.1 to 0.3. (Black to Red) (**Right**) SGD with different contrast loss strengthen (0.0, 0.1, 0.01). Through out the experiments, we find uniform shifting behavior for different algorithm with different strength but the relationship between $\max_i \lambda_{\max}(H_i)$ and $\lambda_{\max}(S)$ form strong regression line.

## C.3 Experiment details - Local Linear stability in quadratic loss for different algorithms $(B, \sigma)$

The experiments in this section serve to understand the local behavior of in terms of the linear stability. By studying the behavior near the local minimum, we aim to verify the correctness of our theory. We follow the same experiment set up to reproduce the plot in Dexter et al. [2024]. We first initialize $H_i = me_1e_1^T$ for all $i \in [\sigma]$ and $H_i = me_{i-\sigma+1}e_{i-\sigma+1}^T$ otherwise. We use $m = \frac{2n}{\sigma}$ so that the sharpness of the minima ($\lambda_{\max}(H)$) is controlled to be 2. We set the learning rate to be smaller than 1 make sure diverging behavior arise due to noise not the sharpness. The loss function in the optimization is $l(w) = \frac{1}{n}\sum_{i=1}^{n} wH_iw$ and the gradient is $\nabla l(w) = \frac{2}{n}\sum_{i=1}^{n} H_iw$ that satisfies our theory setting. For all the experiment in this section, we set $n = 100$. For each set of parameters $(B, \eta, \sigma)$, we determine divergence or converence by conducting 1000 steps update of the weight and calculate the norm of the weight. If the weight norm is 1000 times larger than original initialization, we classify it as diverging and vice versa. For each tuple, we perform the experiment 10 times. If the diverging behavior occurs more than half of the experiments set, we mark the specific tuple as diverging. The experiments involved in our work are done with CPU only.

## C.4 Experiment details - Local Linear stability in mse loss for different algorithms in 2-layer ReLU network.

We use the dimension of data $d = 100$ and the dimension of hidden layer is set to 50. Further, we use the batch size $B = 10$, the SAM $\rho = 0.01$, and the learning rate $\eta = 0.01$. We train for 50 epochs and log the loss over epochs. All experiments comparing different algorithms are done with same initialization using the same random seed. The results are averaged over 5 runs.

## C.5 Experiment details - Global: The role of coherence in the training.

To make the analysis more computationally tractable while tracking multiple quantities simultaneously, we reduce the model size: the input dimension is set to 15, the hidden layer size to 10, and the number of training samples to 50. All other hyperparameters remain the same as in the Section 4.

## C.6 Some identities and definition

We summarize the background and identities used through out the proof.

**Definition 3.** *The definition of Hessian and subset of Hessian where $x_i$ is random variables with Bernoulli distribution*

$$H_t = \frac{1}{B} \sum_{i=1}^n x_i H_i , \quad H = \frac{1}{n} \sum_{i=1}^n H_i \tag{10}$$

**Lemma C.1.** *Consider two matrix A, B with A being Positive semidefinite, then*

$$\lambda_{\max}(A) \operatorname{Tr}[B] \geq \operatorname{Tr}[AB] \geq \lambda_{\min}(A) \operatorname{Tr}[B] \tag{11}$$

*The $\lambda_{\min}, \lambda_{\max}$ are smallest and largest eigenvalue of the matrix A.*

**Lemma C.2.** *Consider two matrix A, B, C, then*

$$\operatorname{Tr}[ABC] = \operatorname{Tr}[BCA] = \operatorname{Tr}[CAB] \tag{12}$$

**Lemma C.3.** $l_1$-$l_2$ *norm inequality: For any $x \in \mathbb{R}, ||x||_2 \leq ||x||_1 \leq \sqrt{d}||x||_2$*

**Lemma C.4.** ***Binomial coefficient:*** *For all $n, k \in \mathbb{N}$ such that $k \leq n$, the binomial coefficients satisfy that*

$$\binom{n}{k} = \binom{n-1}{k-1} + \binom{n-1}{k} \tag{13}$$

**Lemma C.5.** *For any matrix $M \in \mathbb{R}^{n \times n}$, $||M||_F \leq ||M||_{S_1} \leq \sqrt{n}||M||_F$, where $||M||_{S_p}$ is p norm of the spectrum of M, and the inequality is obtain through $l_1$-$l_2$ norm inequality.*

**Lemma C.6.** *For matrices $M_1...M_k \in \mathbb{R}^{n \times n}$, $\operatorname{Tr}[M_1...M_k] \leq ||M_1...M_k||_{S_1}$ (see Bhatia [1997])*

## C.7 Proof for Random perturbation

**Theorem C.7.** *Give update rule (3),*

1. *The condition for divergence is the same as that for SGD [Dexter et al., 2024] as follows:*

$$\eta \geq \frac{\sigma}{\lambda_1}(\frac{n}{b} - 1)^{-\frac{1}{2}}$$

2. *(Comparative Divergence Speed) Suppose $\operatorname{Tr}[J^{2k}] \leq C_0 \alpha^k$ for some constants $C_0$ and $\alpha_k$, then the divergence rate of the random perturbation method is asymptotically within a constant factor of that of standard SGD:*

$$\lim_{k \to \infty} \frac{E[||w_k||^2]_{Random, \ lower \ bound}}{E[||w_k||^2]_{SGD, \ lower \ bound}} = \mathcal{O}(1)$$

3. *Suppose the step size satisfies the convergence criterion established in prior stability analyses (e.g., Dexter et al. [2024]). Then, under the random perturbation update (3), the expected squared norm of the iterates remains bounded as $k \to \infty$:*

$$\lim_{k \to \infty} E[w_k^T w_k]_{upper \ bound} = \mathcal{O}(1)$$

*Proof.* Define $H = \frac{1}{n} \sum_{i=1}^n H_i$. Now consider k steps after, we can have expression for $w_k$ as following:

$$w_k = \hat{J}_k...\hat{J}_1 w_0 - \eta \sum_{t=1}^k (\prod_{t'=t+1}^k \hat{J}_{t'}) H_t \delta_t \tag{14}$$

We consider the dot product of $w_k$ and take expectation over all random process in between:

$$E[w_k^T w_k] = E[(\hat{J}_k...\hat{J}_1 w_0 - \eta \sum_{t=1}^k (\prod_{t'=t+1}^k \hat{J}_{t'}) H_t \delta_t)^T (\hat{J}_k...\hat{J}_1 w_0 - \eta \sum_{t=1}^k (\prod_{t'=t+1}^k \hat{J}_{t'}) H_t \delta_t)]$$

$$= E[w_0^T \hat{J}_1...\hat{J}_k \hat{J}_k...\hat{J}_1 w_0] + \eta^2 E[(\sum_{t=1}^k (\prod_{t'=t+1}^k \hat{J}_{t'}) H_t \delta_t))^T (\sum_{t=1}^k (\prod_{t'=t+1}^k \hat{J}_{t'}) H_t \delta_t))] \tag{15}$$

We first consider the second term. Note that all the cross terms are eliminated as they are independent to each other:

$$\eta^2 E[(\sum_{t=1}^{k}(\prod_{t'=t+1}^{k}\hat{J}_{t'})H_t\delta_t))^T(\sum_{t=1}^{k}(\prod_{t'=t+1}^{k}\hat{J}_{t'})H_t\delta_t))] = \eta^2 E[\sum_{i=1}^{k}\delta_t^T H_t(\hat{J}_{t+1}...\hat{J}_K^2...\hat{J}_{t+1})H_t\delta_t]$$

$$= \eta^2 E[\sum_{i=1}^{k}\mathrm{Tr}((\hat{J}_{t+1}...\hat{J}_K^2...\hat{J}_{t+1})(H_t\delta_t\delta_t^T H_t)]$$

$$= \eta^2\sigma_1^2\sum_{i=1}^{k}E[\mathrm{Tr}(\hat{J}_{t+1}...\hat{J}_K^2...\hat{J}_{t+1})]E[H_t^2]$$

$$\geq \eta^2\sigma_1^2\lambda_{\min}^2\sum_{i=1}^{k}E[\mathrm{Tr}((\hat{J}_{t+1}...\hat{J}_K^2...\hat{J}_{t+1}))] \tag{16}$$

The term $\mathrm{Tr}((J_{t+1}...J_K^2...J_{t+1}))$ can be decomposed into the following according to Dexter et al. [2024].

$$\eta^2\sigma_1^2\lambda_{\min}^2(H)\sum_{t=1}^{k}E[\mathrm{Tr}((J_{t+1}...J_K^2...J_{t+1}))] \geq \eta^2\sigma_1^2\lambda_{\min}^2(H)(\sum_{t=1}^{k}\mathrm{Tr}[J^{2t} + \eta^{2t}(\frac{1}{Bn} - \frac{1}{n^2})^t\sum_{y_1...y_t=1}^{n}H_{y_1}...H_{y_t}^2...H_{y_1}]$$

$$\geq \eta^2\sigma_1^2\lambda_{\min}^2(H)(\sum_{t=1}^{k}\mathrm{Tr}[J^{2t}] + (\frac{\eta}{\sigma})^{2t}(\frac{n}{b} - 1)^t\lambda_{\max}(H)^{2t}) \tag{17}$$

The last term represent the growth of the perturbation over time step. Contrary to the original analysis, the dependency of the magnitude is a summation of geometric series. However, despite the summation dependency, the criterion for diverging is still the same as we only require that $(\frac{\eta}{\sigma})^2(\frac{n}{b} - 1)\lambda_{\max}(H)^2$ to be larger than 1. i.e.,

$$\eta \geq \frac{\sigma}{\lambda_{\max}}(\frac{n}{b} - 1)^{-\frac{1}{2}} \tag{18}$$

Now, observe that the perturbation base method will not change the fundamental criterion for the diverging. However, it will change the speed of diverging. We first consider the summation.

$$\eta^2\sigma_1^2\lambda_{\min}^2(H)\sum_{t=1}^{k}(\frac{\eta}{\sigma})^{2t}(\frac{n}{b} - 1)^t\lambda_{\max}^{2t} = \eta^2\sigma_1^2\lambda_{\min}^2\frac{(\frac{\eta}{\sigma}\lambda_{\max})^2(\frac{n}{b} - 1)((\frac{\eta}{\sigma}\lambda_{\max})^{2k}(\frac{n}{b} - 1)^k - 1)}{(\frac{\eta}{\sigma}\lambda_{\max})^2(\frac{n}{b} - 1) - 1} \tag{19}$$

Now we impose assumption on the growth of the $\mathrm{Tr}(J^{2k})$ by assuming that it grow with pattern $C_0\alpha^k$ and calculate the sum of it.

$$\sum_{t=1}^{k}C_0\alpha^t = \frac{C_0\alpha(\alpha^k - 1)}{\alpha - 1} \tag{20}$$

Finally, we temporarily denote the term $(\frac{\eta}{\sigma}\lambda_{\max}(H))^2(\frac{n}{b} - 1)$ to be $r$ and the overall lower bound for the calculation will be:

$$E[w_k^T w_k] \geq C_0\alpha^k + \frac{1}{nd^5}r^k + \eta^2\sigma_1^2\lambda_{\min}^2(\frac{C_0\alpha(\alpha^k - 1)}{\alpha - 1} + \frac{1}{nd^5}\frac{r(r^k - 1)}{r - 1}) \tag{21}$$

Now we set the $\sigma_1 = 1$ and compare with the original naive SGD and set that $\alpha \geq r$

$$\lim_{k \to \infty} \frac{E[w_k^T w_k]_{\text{Random, lower bound}}}{E[w_k^T w_k]_{\text{SGD, lower bound}}} = 1 + \eta^2 \sigma_1^2 \lambda_{\min}^2(H) \frac{\alpha}{\alpha - 1} \tag{22}$$

The escaping speed of the random perturbation base method is faster by a constant. Now we set that $\alpha \leq r$

$$\lim_{k \to \infty} \frac{E[w_k^T w_k]_{\text{Random, lower bound}}}{E[w_k^T w_k]_{\text{SGD, lower bound}}} = 1 + \eta^2 \sigma_1^2 \lambda_{\min}^2 \frac{r}{r - 1} \tag{23}$$

No matter which term dominates, we will have constant faster escaping efficiency compared to the original naive SGD.

Now, we consider the convergence behavior, we first note that there exists $\epsilon$ and $C$ such that $E[\hat{J}_k...\hat{J}_1^2...\hat{J}_k] \leq C((1 - \epsilon)^2 + \epsilon)^k$. Here, we temporarily denote $((1 - \epsilon)^2 + \epsilon)$ to be $r$. We apply this identity to the above equation and we will get the following:

$$E[w_k^T w_k] \leq Cr^k + \eta^2 \lambda_{\max} \sum_{t=1}^{k} Cr^t$$

$$= Cr^k + \eta^2 \lambda_{\max} \frac{r(1 - r^k)}{1 - r} \tag{24}$$

We consider long term behavior (i.e., $k \to \infty$)

$$\lim_{k \to \infty} E[w_k^T w_k] \leq \eta^2 \lambda_{\max} \frac{r}{1 - r} \tag{25}$$

We observe that there exist residual terms relating to the perturbation itself and this fits our intuition that the random perturbation method will usually hover around the minimum as the noise injected can lead to less accurate estimation of the gradient direction. $\square$

### C.8 Proof for divergence theorem

**Theorem C.8.** *For update rules as following:*

$$W_{t+1} = (I - \eta H_t (I + \frac{\rho}{\alpha} H)) W_t \tag{26}$$

*Define $\hat{J}_t = (I - \eta H_t (I + \frac{\rho}{\alpha} H))$, then*

**1.There exist $M_k$ such that**

$$E[\hat{J}_k^T ... \hat{J}_1^T \hat{J}_1 ... \hat{J}_k] \succeq M_k \tag{27}$$

*with*

$$M_k = J^{2k} + \eta^{2k} (\frac{1}{Bn} - \frac{1}{n^2})^k \sum_{y_1 ... y_k = 1}^{n} (I + \frac{\rho}{\alpha} H) H_{y_k} ... (I + \frac{\rho}{\alpha} H) H_{y_1}^2 (I + \frac{\rho}{\alpha} H) ... H_{y_k} (I + \frac{\rho}{\alpha} H) \tag{28}$$

**2.The Trace of $M_k$ can be lower bounded by the following:**

$$\mathrm{Tr}[M_k] \geq \eta^{2k} (\frac{1}{Bn} - \frac{1}{n^2})^k (1 + \frac{\rho}{\alpha} \lambda_{\min}(H))^{2k} \mathrm{Tr}[\sum_{y_1 ... y_k = 1}^{n} H_{y_k} ... H_{y_1}^2 ... H_{y_k}] \tag{29}$$

**3. The Trace of $M_k$ is lower bounded through $\sigma$:**

$$\mathrm{Tr}[M_k] \geq \eta^{2k} (\frac{n}{B} - 1)^k (1 + \frac{\rho}{\alpha} \lambda_{\min}(H))^{2k} \frac{1}{\sigma^{2k}} \frac{1}{nd^5} \lambda_{\max}(H)^{2k} \tag{30}$$

**4. The diverging criterion for SAM under linear stability is:**

$$\lambda_{\max}(H) \geq \frac{\sigma}{\eta} (\frac{n}{B} - 1)^{-\frac{1}{2}} (1 + \frac{\rho}{\alpha} \lambda_{\min}(H))^{-1} \tag{31}$$

*Proof.* We prove by induction as follows.

**Base case: k=1**

$$E[\hat{J}_1^T \hat{J}_1] = E[(I - \eta H_t (I + \frac{\rho}{\alpha}) H)^T (I - \eta H_t (I + \frac{\rho}{\alpha}) H)]$$

$$= E[I - 2\eta H_t - \frac{\eta \rho}{\alpha} (H_t H + H H_t) + \eta^2 H_t^2 + \frac{\eta^2 \rho}{\alpha} (H_t^2 H + H H_t^2) + \frac{\eta^2 \rho^2}{\alpha^2} H H_t^2 H]$$

$$= I - 2\eta H - 2\frac{\eta \rho}{\alpha} H^2 + \eta^2 E[H_t^2] + \frac{\eta^2 \rho}{\alpha} (E[H_t^2] H + H E[H_t^2]) + \frac{\eta^2 \rho^2}{\alpha^2} H E[H_t^2] H \tag{32}$$

We know that

$$E[H_t^2] = H^2 + (\frac{1}{Bn} - \frac{1}{n^2}) \sum_{i=1} H_i^2 \tag{33}$$

and we will have

$$E[\hat{J}_1^T \hat{J}_1] = J^2 + \eta^2 (\frac{1}{Bn} - \frac{1}{n^2})(I + \frac{\rho}{\alpha} H)(\sum_{i=1} H_i^2)(I + \frac{\rho}{\alpha} H) \tag{34}$$

$$= M_1$$

**Induction case: k-1 to k**

$$E[\hat{J}_k^T...\hat{J}_1^T\hat{J}_1...\hat{J}_k] \succeq E[\hat{J}_k^T M_{k-1}\hat{J}_k]$$

$$= E[(I - \eta H_k - \frac{\eta\rho}{\alpha}H_k H)^T M_{k-1}(I - \eta H_k - \frac{\eta\rho}{\alpha}H_k H)]$$

$$= E[M_{k-1} - \eta M_{k-1}H_k - \frac{\eta\rho}{\alpha}M_{k-1}H_k H - \eta H_k M_{k-1} + \eta^2 H_k M_{k-1}H_k + \frac{\eta^2\rho}{\alpha}H_k M_{k-1}H_k H -$$

$$\frac{\eta\rho}{\alpha}HH_k M_{k-1} + \frac{\eta^2\rho}{\alpha}HH_k M_{k-1}H_k + \frac{\eta^2\rho^2}{\alpha^2}HH_k M_{k-1}H_k H]$$

$$= JM_{k-1}J + \eta^2(\frac{1}{nB} - \frac{1}{n^2})(I + \frac{\rho}{\alpha}H)(\sum_i H_i M_{k-1}H_i)(I + \frac{\rho}{\alpha}H)$$

(35)

Now, we subsititude the expression of $M_{k-1}$ into the expression and we will have the follows:

$$E[\hat{J}_k^T...\hat{J}_1^T\hat{J}_1...\hat{J}_k] \succeq JM_{k-1}J + \eta^2(\frac{1}{nB} - \frac{1}{n^2})(I + \frac{\rho}{\alpha}H)(\sum_i H_i M_{k-1}H_i)(I + \frac{\rho}{\alpha}H)$$

$$= J[J^{2(k-1)} + \eta^{2(k-1)}(\frac{1}{Bn} - \frac{1}{n^2})^{(k-1)}\sum_{y_1...y_k=1}^n (I + \frac{\rho}{\alpha}H)H_{y_{k-1}}...(I + \frac{\rho}{\alpha}H)H_{y_1}^2(I + \frac{\rho}{\alpha}H)...H_{y_{k-1}}(I + \frac{\rho}{\alpha}H)]J +$$

$$\eta^2(\frac{1}{nB} - \frac{1}{n^2})(I + \frac{\rho}{\alpha}H)(\sum_i H_i[J^{2(k-1)} +$$

$$\eta^{2(k-1)}(\frac{1}{Bn} - \frac{1}{n^2})^{(k-1)}\sum_{y_1...y_k=1}^n (I + \frac{\rho}{\alpha}H)H_{y_{k-1}}...(I + \frac{\rho}{\alpha}H)H_{y_1}^2(I + \frac{\rho}{\alpha}H)...H_{y_{k-1}}(I + \frac{\rho}{\alpha}H)]H_i)(I + \frac{\rho}{\alpha}H)$$

$$\succeq J^{2k} + \eta^{2k}(\frac{1}{Bn} - \frac{1}{n^2})^k \sum_{y_1...y_k=1}^n (I + \frac{\rho}{\alpha}H)H_{y_k}...(I + \frac{\rho}{\alpha}H)H_{y_1}^2(I + \frac{\rho}{\alpha}H)...H_{y_k}(I + \frac{\rho}{\alpha}H)$$

$$= M_k$$

(36)

Now, we wish to analyze the trace of the matrix $M_k$. For simplicity, we analyze the latter term of the expression.

$$\text{Tr}[M_k] = \text{Tr}[\eta^{2k}(\frac{1}{Bn} - \frac{1}{n^2})^k \sum_{y_1...y_k=1}^n (I + \frac{\rho}{\alpha}H)H_{y_k}...(I + \frac{\rho}{\alpha}H)H_{y_1}^2(I + \frac{\rho}{\alpha}H)...H_{y_k}(I + \frac{\rho}{\alpha}H)]$$

(37)

We first focus on specific term in the summation.

$$\text{Tr}[(I + \frac{\rho}{\alpha}H)H_{y_k}...(I + \frac{\rho}{\alpha}H)H_{y_1}^2(I + \frac{\rho}{\alpha}H)...H_{y_k}(I + \frac{\rho}{\alpha}H)] =$$

$$\text{Tr}[(I + \frac{\rho}{\alpha}H)^2 H_{y_k}(I + \frac{\rho}{\alpha}H)H_{y_{k-1}}...(I + \frac{\rho}{\alpha}H)H_{y_1}^2(I + \frac{\rho}{\alpha}H)...H_{y_{k-1}}(I + \frac{\rho}{\alpha}H)H_{y_k}]$$

$$\geq (1 + \frac{\rho}{\alpha}\lambda_{\min}(H))^2 \text{Tr}[H_{y_k}...(I + \frac{\rho}{\alpha}H)H_{y_1}^2(I + \frac{\rho}{\alpha}H)...H_{y_k}]$$

$$= (1 + \frac{\rho}{\alpha}\lambda_{\min}(H))^3 \text{Tr}[H_{y_{k-1}}...(I + \frac{\rho}{\alpha}H)H_{y_1}^2(I + \frac{\rho}{\alpha}H)...H_{y_{k-1}}(I + \frac{\rho}{\alpha}H)H_{y_k}^2]$$

$$\geq (1 + \frac{\rho}{\alpha}\lambda_{\min}(H))^4 \text{Tr}[H_{y_{k-1}}H_{y_k}^2 H_{y_{k-1}}(I + \frac{\rho}{\alpha}H)...(I + \frac{\rho}{\alpha}H)H_{y_1}^2(I + \frac{\rho}{\alpha}H)...(I + \frac{\rho}{\alpha}H)]$$

(38)

Here, we use the lemma C.1. By continue pruning, we will have the following:

$$\text{Tr}[(I + \frac{\rho}{\alpha}H)H_{y_k}...(I + \frac{\rho}{\alpha}H)H_{y_1}^2(I + \frac{\rho}{\alpha}H)...H_{y_k}(I + \frac{\rho}{\alpha}H)] \geq (1 + \frac{\rho}{\alpha}\lambda_{\min}(H))^{2k}\text{Tr}[H_{y_k}...H_{y_1}^2...H_{y_k}]$$

(39)

We apply this to the summation in $M_k$ and we will get

$$\text{Tr}[M_k] \geq \eta^{2k}(\frac{1}{Bn} - \frac{1}{n^2})^k(1 + \frac{\rho}{\alpha}\lambda_{\min}(H))^{2k}\text{Tr}[\sum_{y_1...y_k=1}^{n} H_{y_k}...H_{y_1}^2...H_{y_k}]$$

(40)

Now, we connect the $M_k$ with the coherence measure in the following:

$$
\begin{aligned}
\text{Tr}[M_k] &\geq \eta^{2k}(\frac{1}{Bn} - \frac{1}{n^2})^k(1 + \frac{\rho}{\alpha}\lambda_{\min}(H))^{2k}\sum_{y_1...y_k=1}^{n}\text{Tr}[H_{y_k}...H_{y_1}^2...H_{y_k}] \\
&= \eta^{2k}(\frac{1}{Bn} - \frac{1}{n^2})^k(1 + \frac{\rho}{\alpha}\lambda_{\min}(H))^{2k}\sum_{y_1...y_k=1}^{n}||H_{y_k}...H_{y_1}||_F^2 \\
&\geq \eta^{2k}(\frac{1}{Bn} - \frac{1}{n^2})^k(1 + \frac{\rho}{\alpha}\lambda_{\min}(H))^{2k}\sum_{y_1...y_k=1}^{n}\frac{1}{d}||H_{y_k}...H_{y_1}||_{S_1}^2 \\
&\geq \eta^{2k}(\frac{1}{Bn} - \frac{1}{n^2})^k(1 + \frac{\rho}{\alpha}\lambda_{\min}(H))^{2k}\sum_{y_1...y_k=1}^{n}\frac{1}{d}\text{Tr}[H_{y_k}...H_{y_1}]^2 \\
&\geq \eta^{2k}(\frac{1}{Bn} - \frac{1}{n^2})^k(1 + \frac{\rho}{\alpha}\lambda_{\min}(H))^{2k}\sum_{y=1}^{n}\frac{1}{d}\text{Tr}[H_y^k]^2 \\
&\geq \eta^{2k}(\frac{1}{Bn} - \frac{1}{n^2})^k(1 + \frac{\rho}{\alpha}\lambda_{\min}(H))^{2k}\frac{1}{nd}(\sum_{y=1}^{n}\text{Tr}[H_y^k])^2
\end{aligned}
$$

(41)

for the above we use lemma C.5 and we know the following from Dexter et al. [2024]:

$$\frac{n^k}{d^2\sigma^k}\text{Tr}[H^k] \leq \sum_{y=1}^{n}\text{Tr}[H_y^k]$$

(42)

Finally, we can have the following:

$$
\begin{aligned}
\text{Tr}[M_k] &\geq \eta^{2k}(\frac{1}{Bn} - \frac{1}{n^2})^k(1 + \frac{\rho}{\alpha}\lambda_{\min}(H))^{2k}\frac{1}{nd}(\frac{n^k}{d^2\sigma^k})^2(\text{Tr}[H^k])^2 \\
&= \eta^{2k}(\frac{1}{Bn} - \frac{1}{n^2})^k(1 + \frac{\rho}{\alpha}\lambda_{\min}(H))^{2k}\frac{1}{nd}(\frac{n^k}{d^2\sigma^k})^2(\text{Tr}[H^k])^2 \\
&= \eta^{2k}(\frac{n}{B} - 1)^k(1 + \frac{\rho}{\alpha}\lambda_{\min}(H))^{2k}\frac{1}{\sigma^{2k}}\frac{1}{nd^5}(\text{Tr}[H^k])^2 \\
&\geq \eta^{2k}(\frac{n}{B} - 1)^k(1 + \frac{\rho}{\alpha}\lambda_{\min}(H))^{2k}\frac{1}{\sigma^{2k}}\frac{1}{nd^5}\lambda_{\max}(H)^{2k}
\end{aligned}
$$

(43)

We then will have the following condition for diverging:

$$\lambda_{\max}(H) \geq \frac{\sigma}{\eta}(\frac{n}{B} - 1)^{-\frac{1}{2}}(1 + \frac{\rho}{\alpha}\lambda_{\min}(H))^{-1}$$

(44)

$\square$

## C.9  Proof for convergence theorem

**Theorem C.9.** *For update rules as following:*

$$W_{t+1} = (I - \eta H_t(I + \frac{\rho}{\alpha}H))W_t \tag{45}$$

*1. There exist $N_r$ such that*

$$E[\hat{J}_k^T...\hat{J}_1^T \hat{J}_1...\hat{J}_k] \preceq \sum_{r=0}^{k}(1-\epsilon)^{2(k-r)}\binom{k}{r}N_r \tag{46}$$

*and*

$$N_k = \eta^{2k}(\frac{1}{nB} - \frac{1}{n^2})^k \sum_{y_1,...y_r=1}^{n}(I + \frac{\rho}{\alpha}H)H_{y_k}...(I + \frac{\rho}{\alpha}H)H_{y_1}^2(I + \frac{\rho}{\alpha}H)...H_{y_k}(I + \frac{\rho}{\alpha}H) \tag{47}$$

*2. The $N_r$ can be upper bounded as following*

$$\text{Tr}[N_r] \leq \eta^{2k}(\frac{1}{B} - \frac{1}{n})^k d^{3k+\frac{1}{2}} n^{4k}\frac{\lambda_{\max}(H_{SAM})^{4k}}{\sigma_{SAM}^{2k}} \tag{48}$$

*3. Suppose there exist $\epsilon \in (0,1)$ and we will have converging criterion such that*

$$\frac{\epsilon}{\eta} \leq \lambda_i + \frac{\rho}{\alpha}\lambda_i^2 \leq \frac{2-\epsilon}{\eta} \quad \forall i \in [d] \quad and$$

$$\lim_{k\to\infty}\frac{1}{\epsilon^k}\eta^{2k}(\frac{1}{nB} - \frac{1}{n^2})^k \sum_{y_1,y_2...y_k=1}^{n}(I + \frac{\rho}{\alpha}H)H_{y_k}...(I + \frac{\rho}{\alpha}H)H_{y_1}^2(I + \frac{\rho}{\alpha}H)...H_{y_k}(I + \frac{\rho}{\alpha}H) = 0 \tag{49}$$

*then we will have that $\lim_{k\to\infty} E[\hat{J}_k^T...\hat{J}_1^T \hat{J}_1...\hat{J}_k] = 0$*

*Proof.* We first define $N_r$ as follows:

$$N_k = \eta^{2k}(\frac{1}{nB} - \frac{1}{n^2})^k \sum_{y_1,...y_r=1}^{n}(I + \frac{\rho}{\alpha}H)H_{y_k}...(I + \frac{\rho}{\alpha}H)H_{y_1}^2(I + \frac{\rho}{\alpha}H)...H_{y_k}(I + \frac{\rho}{\alpha}H) \tag{50}$$

We define $N_0 = I$

we want to prove the following:

$$E[\hat{J}_k^T...\hat{J}_1^T \hat{J}_1...\hat{J}_k] \preceq \sum_{r=0}^{k}(1-\epsilon)^{2(k-r)}\binom{k}{r}N_r \tag{51}$$

**Base case:  k=1**

$$E[\hat{J}_1^T \hat{J}_1] = E[(I - \eta\hat{H} - \frac{\eta\rho}{\alpha}\hat{H}H)^T(I - \eta\hat{H} - \frac{\eta\rho}{\alpha}\hat{H}H)]$$

$$= J^2 + \eta^2(\frac{1}{nB} - \frac{1}{n^2})\sum_{i}(I + \frac{\rho}{\alpha}H)H_i^2(I + \frac{\rho}{\alpha}H) \tag{52}$$

$$\preceq (1-\epsilon)^2 N_0 + N_1$$

The first condition $(1-\epsilon)^2 I$ is achieved when the condition of the assumption is satisfied. We demonstrate why that is the case

$$J = I - \eta H(I + \frac{\rho}{\alpha}H) \tag{53}$$

and observe the following

$$-(1 - \epsilon)^2 I \preceq J^2 \preceq (1 - \epsilon)^2 I \tag{54}$$

First, we focus on the

$$J^2 \preceq (1 - \epsilon)^2 I \tag{55}$$

By replacing the definition of $J$ into the equation, we will reach

$$(I - \eta H(I + \frac{\rho}{\alpha} H))^2 \preceq (1 - \epsilon)^2 I \tag{56}$$

By removing the square term,

$$(I - \eta H(I + \frac{\rho}{\alpha} H)) \preceq (1 - \epsilon) I \tag{57}$$

Rearrange will give

$$\epsilon I \preceq \eta H(I + \frac{\rho}{\alpha} H) \tag{58}$$

By decomposing each eigenvalue direction, we can have that

$$\frac{\epsilon}{\eta} \leq \lambda_i + \frac{\rho}{\alpha} \lambda_i^2 \tag{59}$$

We perform the same operation on the other direction and we will have

$$\lambda_i + \frac{\rho}{\alpha} \lambda_i^2 \leq \frac{2 - \epsilon}{\eta} \tag{60}$$

The $\epsilon$ in our analysis represent the deterministic term in each step as we use it to upper bound $J^2$. Compared to SGD, the $\epsilon$ can be larger as the term $I - \eta H$ is larger than $I - \eta H - \eta \frac{\rho}{\alpha} H^2$ in each direction. The deterministic part of update process can shrink faster compared to the SGD. The analysis implicitly incorporate it through $\epsilon$. For the other part, $N_1$ represent the randomness in the operation which can be directly checked that if we increase the batch size to $n$, we will have the term vanished. The format of the $N_r$ rely on how different sample align with each other and this form origin of the noise. Compared to the tradition analysis, we can have more subtle observation of the noise through this specific form as we no longer need to assume the structure of the noise or we can assume the how each sample align with each other to see the final form of the noise and this can reveal more insight of the relationship between sample for further analysis. Now, we proceed to induction step.

**Induction case: k-1**

$$E[\hat{J}_k^T ... \hat{J}_1^T \hat{J}_1 ... \hat{J}_k] \preceq E[\hat{J}_k^T (\sum_{r=0}^{k-1} (1 - \epsilon)^{2(k-1-r)} \binom{k-1}{r} N_r) \hat{J}_k]$$

$$= J^T (\sum_{r=0}^{k-1} (1 - \epsilon)^{2(k-1-r)} \binom{k-1}{r} N_r) J + \eta^2 (\frac{1}{nb} - \frac{1}{n^2}) \sum_i (I + \frac{\rho}{\alpha} H) H_i (\sum_{r=0}^{k-1} (1 - \epsilon)^{2(k-1-r)} \binom{k-1}{r} N_r) H_i (I + \frac{\rho}{\alpha} H)$$

$$\preceq (1 - \epsilon)^2 \sum_{r=0}^{k-1} (1 - \epsilon)^{2(k-1-r)} \binom{k-1}{r} N_r + \eta^2 (\frac{1}{nb} - \frac{1}{n^2}) \sum_i \sum_{r=0}^{k-1} (1 - \epsilon)^{2(k-1-r)} \binom{k-1}{r} (I + \frac{\rho}{\alpha} H) H_i N_r H_i (I + \frac{\rho}{\alpha} H)$$

$$= \sum_{r=0}^{k-1} (1 - \epsilon)^{2(k-r)} \binom{k-1}{r} N_r + \eta^2 (\frac{1}{nb} - \frac{1}{n^2}) \sum_{r=0}^{k-1} (1 - \epsilon)^{2(k-1-r)} \binom{k-1}{r} \sum_i (I + \frac{\rho}{\alpha} H) H_i N_r H_i (I + \frac{\rho}{\alpha} H)$$

$$= \sum_{r=0}^{k-1} (1 - \epsilon)^{2(k-r)} \binom{k-1}{r} N_r + \sum_{r=0}^{k-1} (1 - \epsilon)^{2(k-1-r)} \binom{k-1}{r} N_{r+1}$$

$$\tag{61}$$

By reordering the term, we will have the following using lemma C.4:

$$\sum_{r=0}^{k-1}(1-\epsilon)^{2(k-r)}\binom{k-1}{r}N_r + \sum_{r=1}^{k}(1-\epsilon)^{2(k-r)}\binom{k-1}{r-1}N_r$$

$$= (1-\epsilon)^{2k}N_0 + \sum_{r=0}^{k-1}((1-\epsilon)^{2(k-r)}\binom{k-1}{r} + (1-\epsilon)^{2(k-r)}\binom{k-1}{r-1})N_r + N_k \qquad (62)$$

$$= \sum_{r=0}^{k}(1-\epsilon)^{2(k-r)}\binom{k}{r}N_r$$

Similarly, as we require that the $\mathrm{Tr}[N_r]$ term to be smaller than $\epsilon^r$, we can upper bound the term with constant $C$ such that $\frac{1}{\epsilon^r}\mathrm{Tr}[N_r] \le C$, and therefore,

$$\sum_{r=0}^{k}(1-\epsilon)^{2(k-r)}\binom{k}{r}\mathrm{Tr}[N_r] \le C\sum_{r=0}^{k}\binom{k}{r}(1-\epsilon)^{2(k-r)}\epsilon^r$$

$$= C((1-\epsilon)^2 + \epsilon)^k \qquad (63)$$

For the last step, as we ask the $\mathrm{Tr}[N_r]$ term to be smaller than $\epsilon$, we can further bound it. Despite we can upper bound it through $\epsilon$, we can still analysis its magnitude. If it is smaller, it will also converge faster. The distinction between SAM and SGD is that SAM has extra multiplication $(I + \frac{\rho}{\alpha}H)$ and this result from the operation with the gradient ascent intermediate step. We can find that this can potentially make the process unstable as it amplify the noise through out the process and this fit in to our general believe that SAM can make the optimization process less stable but still converge fast due to the shrink of the deterministic term. Note that unlike the tradition analysis that require step size to be $\eta \le \frac{2}{\lambda_{\max}(H)}$, we ask for different criterion for converging. This is due to the fact that we analysis the origin of noise which comes from alignment of samples and the traditional analysis focus more on the deterministic part which directly involve eigenvalue of Hessian and usually the analysis specify the noise with covariance matrix instead. □

To answer the relationship between $\epsilon$ and the $N_r$. We first consider the following inequility

$$\lambda_1(S)^k \le \mathrm{Tr}(S^k)$$

$$= \sum_{y_1\ldots y_k=1}^{n}||H_{y_1}^{\frac{1}{2}}H_{y_k}^{\frac{1}{2}}||_F\ldots||H_{y_2}^{\frac{1}{2}}H_{y_1}^{\frac{1}{2}}||_F$$

$$= \sum_{y_1\ldots y_k=1}^{n}\mathrm{Tr}(H_{y_1}H_{y_k})\ldots\mathrm{Tr}(H_{y_2}H_{y_1}) \qquad (64)$$

$$= n^{2k}(\mathrm{Tr}(H^2))^k$$

$$\le n^{2k}d^k\lambda_{\max}(H)^{2k}$$

Now, we defined a new form of coherence matrix and Hessian to accomadate the SAM algorithm as following:

$$S_{\mathrm{SAM}_{ij}} = \sqrt{\mathrm{Tr}((I + \frac{\rho}{\alpha}H)H_i(I + \frac{\rho}{\alpha}H)H_j)} \qquad (65)$$

$$H_{\mathrm{SAM}_{ij}} = (I + \frac{\rho}{\alpha}H)\sum_{i=1}^{n}H_i \qquad (66)$$

Now, we go back to the $N_r$

$$\text{Tr}(N_r) = \eta^{2k}(\frac{1}{nB} - \frac{1}{n^2})^k \sum_{y_1...y_k=1}^{n} \text{Tr}((I + \frac{\rho}{\alpha}H)H_{y_k}...(I + \frac{\rho}{\alpha}H)H_{y_1}^2(I + \frac{\rho}{\alpha}H)...H_{y_k}(I + \frac{\rho}{\alpha}H))$$

$$= \eta^{2k}(\frac{1}{nB} - \frac{1}{n^2})^k \sum_{y_1...y_k=1}^{n} ||(I + \frac{\rho}{\alpha}H)H_{y_k}...(I + \frac{\rho}{\alpha}H)H_{y_1}||_F^2$$

$$\leq \eta^{2k}(\frac{1}{nB} - \frac{1}{n^2})^k \sqrt{d} \sum_{y_1...y_k=1}^{n} ||(I + \frac{\rho}{\alpha}H)H_{y_k}||_F^2...||(I + \frac{\rho}{\alpha}H)H_{y_1}||_F^2$$

$$\leq \eta^{2k}(\frac{1}{B} - \frac{1}{n})^k \sqrt{d} \max_{i=1...n} ||(I + \frac{\rho}{\alpha}H)H_i||_F^{2k}$$

$$\leq \eta^{2k}(\frac{1}{B} - \frac{1}{n})^k \sqrt{d} \max_{i=1...n} d^k(\lambda_{\max}((I + \frac{\rho}{\alpha}H)H_i))^{2k}$$

$$\leq \eta^{2k}(\frac{1}{B} - \frac{1}{n})^k d^{3k+\frac{1}{2}} n^{4k} \frac{\max_{i=1...n}(\lambda_{\max}((I + \frac{\rho}{\alpha}H)H_i))^{2k}}{\lambda_{\max}(S_{\text{SAM}})^{2k}} \lambda_{\max}(H_{\text{SAM}})^{4k}$$

$$\leq \eta^{2k}(\frac{1}{B} - \frac{1}{n})^k d^{3k+\frac{1}{2}} n^{4k} \frac{\lambda_{\max}(H_{\text{SAM}})^{4k}}{\sigma_{\text{SAM}}^{2k}}$$

$$(67)$$

In our analysis, through new definition of the coherence matrix and Hessian, we find that SAM is performing optimization on the loss surface that is amplified by $I + \frac{\rho}{\alpha}H$. The loss surface is sharper as it give larger eigenvalue in each direction. If we compared about the ratio $\frac{\lambda_{\max}(H)^4}{\sigma^2}$ and $\frac{\lambda_{\max}(H_{\text{SAM}})^4}{\sigma_{\text{SAM}}^2}$, we can see question about which one is larger or smaller will need more information about the exact coherence matrix to determine. They can be the same or different depending on the relationship between samples. However, for both of the method, if the solution give larger coherence measure, they both converge faster for the specific solution and vice versa.

### C.10 Proof for theorem 3.4

*Proof.* We know that the $\nabla f_w(x_i)$ can be written as following:

$$\nabla f_w(x_i) = \begin{bmatrix} \text{ReLU}(W_{1,1}x_i) \\ ... \\ \text{ReLU}(W_{1,d_2}x_i) \\ W_{2,1}\mathbf{1}[W_{1,1}x_i > 0]x_i \\ ... \\ W_{2,j}\mathbf{1}[W_{1,d_2}x_i > 0]x_i \\ W_{2,j}\mathbf{1}[W_{1,1}x_i > 0] \\ ... \\ W_{2,j}\mathbf{1}[W_{1,d_2}x_i > 0] \end{bmatrix} \qquad (68)$$

where the gradient is taken with respect to parameter $(W_2, W_1, b)$ in sequence. For each element in the coherence matrix, we will have

$$S_{i,j} = ||H_i^{\frac{1}{2}}H_j^{\frac{1}{2}}||_F = \sqrt{\text{Tr}(H_j^{\frac{1}{2}}H_i^{\frac{1}{2}}H_i^{\frac{1}{2}}H_j^{\frac{1}{2}})} = \sqrt{\text{Tr}(H_iH_j)} = \sqrt{(\nabla f_w^T(x_i)\nabla f_w(x_j))^2}$$
$$= |\nabla f_w^T(x_i)\nabla f_w(x_j)| \qquad (69)$$

As the activation of the samples are orthogonal to each other in memorizing solution. The orthogonal in activation will also give the gradient orthogonal property and therefore,

$$\text{Tr}(H_iH_j) = (\nabla f_i \nabla f_j)^2 = 0 \qquad (70)$$

Therefore, the coherence matrix is diagonal in the setting. The corresponding coherence measure is small compared to other solution and we can conclude that the memorizing solution is relatively hard to find during optimization process as seen in the prior work with coherence measure. The reverse is also true. If the coherence matrix is diagonal, then the solution is memorizing solution. As if we have two data activation overlap, the gradient product will not be zero. □

## C.11 Proof for theorem 3.5

*Proof.* Suppose we draw a dataset with size n uniformly at random. The coherence matrix becomes block diagonal matrix with eigenvalue being $2(d+1)^{\frac{1}{2}} \max_{i=1...2^C} |S_i|$ where $S_i$ is the set with data matched to specific feature extracted by $W_{1,i}$ and we know the following:

$$S_{i,j} = |\nabla f_w^T(x_i) \nabla f_w(x_j)| = 2(d+1)^{\frac{1}{2}}. \tag{71}$$

We estimate the following:

$$P(\max_{i=1...2^C} |S_i| \geq nu + n\epsilon). \tag{72}$$

We can find that by union bound

$$P(\max_{i=1...2^C} |S_i| \geq nu + n\epsilon) \leq \sum_{i=1}^{2^C} P(|S_i| \geq nu + n\epsilon) \tag{73}$$

Let $u = \frac{1}{2^C}$ and $\epsilon = \sqrt{\frac{C + \log \frac{1}{\delta}}{2n}}$. Also, we know that by $|S_i|$ is a sum of independent variables $X_{ik}$ that fall into the category and we can formulate through chernoff bound:

$$P(|S_i| \geq nu + n\epsilon) = P(\frac{1}{n} \sum_{j=1}^n X_j \geq u + \epsilon) \leq \exp(-2\epsilon n^2) \leq \exp(-(C + \log \frac{1}{\delta})) = e^{-C} \delta \tag{74}$$

Therefore,

$$\begin{aligned} P(\max_{i=1...2^C} |S_i| \geq nu + n\epsilon) &\leq \sum_{i=1}^{2^C} P(|S_i| \geq nu + n\epsilon) \\ &\leq \sum_{i=1}^{2^C} e^{-C} \delta \\ &\leq \delta \end{aligned} \tag{75}$$

$\square$

## C.12 Proof for theorem 3.6

*Proof.* For the generalizing solution, we can analyze the element in the coherence matrix. $\text{Tr}((I + \frac{\rho}{\alpha} H) H_i (I + \frac{\rho}{\alpha} H) H_j)$. We can see that if two samples do not share the same activation, the specific element will be zero. We consider average value for the element that are within the same cluster. As they are in the same cluster, the $H_i = H_j = H_S$

$$\begin{aligned} E[(\sum_k X_k) \sqrt{\text{Tr}[(I + \frac{\rho}{\alpha} H) H_i (I + \frac{\rho}{\alpha} H) H_j]}] &= E[(\sum_k X_k) \sqrt{\text{Tr}[H_i H_j + \frac{\rho}{\alpha} H H_i H_j + \frac{\rho}{\alpha} H_j H H_j + \frac{\rho^2}{\alpha^2} H H_i H H_j]]} \\ &= E[(\sum_k X_k) \sqrt{\text{Tr}[H_S^2] + \frac{2\rho}{\alpha} \text{Tr}[H_S^3] \sum_{k=1}^n X_k + \frac{\rho^2}{\alpha^2} \text{Tr}[H_S^4] \sum_{kk'} X_k X_{k'}}] \\ &= E[(\sum_k X_k) \text{Tr}[H_S] \sqrt{1 + \frac{2\rho}{\alpha} \text{Tr}[H_S] \sum_{k=1}^n X_k + \frac{\rho^2}{\alpha^2} \text{Tr}[H_S^2] \sum_{kk'} X_k X_{k'}}] \end{aligned} \tag{76}$$

where the $\sum_k X_k$ are random variables indicating the sample inside the specific cluster or not. The other term is the strengthen of coherence elementwise. We can observe that this is a convex function in terms of the random variables and therefore we can lower bound it by taking the expectation first in each random variable:

$$E[(\sum_k X_k)\sqrt{\text{Tr}[(I + \frac{\rho}{\alpha}H)H_i(I + \frac{\rho}{\alpha}H)H_j]]}$$

$$\geq E[\sum_k X_k]\text{Tr}[H_S]\sqrt{1 + \frac{1}{n}\frac{2\rho}{\alpha}\text{Tr}[H_S]E[\sum_{k=1}^n X_k] + \frac{1}{n^2}\frac{\rho^2}{\alpha^2}\text{Tr}[H_S^2]E[\sum_{kk'} X_k X_{k'}]} \tag{77}$$

The key lies in the term $E[\sum_{kk'} X_k X_{k'}]$ which is not simply the multiplication of the two individual probability and we an find that it is $\frac{1}{2^{2c}} + \frac{1}{n}(\frac{1}{2^c} - \frac{1}{2^{2c}})$ and we will have the following:

$$E[(\sum_k X_k)\sqrt{\text{Tr}[(I + \frac{\rho}{\alpha}H)H_i(I + \frac{\rho}{\alpha}H)H_j]}$$

$$\geq \frac{n}{2^c}2(d+1)^{\frac{1}{2}}\sqrt{1 + \frac{2\rho}{\alpha}\frac{1}{2^c}2(d+1)^{\frac{1}{2}} + \frac{\rho^2}{\alpha^2}(\frac{1}{2^{2c}} + \frac{1}{n}(\frac{1}{2^c} - \frac{1}{2^{2c}}))4(d+1)} \tag{78}$$

$$= \frac{n}{2^c}2(d+1)^{\frac{1}{2}}\sqrt{(1 + \frac{\rho}{\alpha}\frac{2(d+1)^{\frac{1}{2}}}{2^c})^2 + \frac{\rho^2}{\alpha^2}(\frac{1}{n}(\frac{1}{2^c} - \frac{1}{2^{2c}}))4(d+1)}$$

Now, the even stronger dependency of the number of features used can also be translated to the probability statement. With probability $1 - \delta$, the eigenvalue of the coherence matrix is upper bounded by $\mathcal{O}(\frac{n}{2^c}(d+1)^{\frac{1}{2}}\sqrt{(1 + \frac{\rho}{\alpha}\frac{2(d+1)^{\frac{1}{2}}}{2^c})^2 + \frac{\rho^2}{\alpha^2}(\frac{1}{n}(\frac{1}{2^c} - \frac{1}{2^{2c}}))4(d+1)})$ using the same method as in appendix C.12.

The additional higher order interacting term give strong additional bias toward solution with lower C, by observation, we can see that the term become more significant when the data dimension (also model dimension) becomes higher and cannot be neglect for modern deep learning scenario in terms of overparameter region. Also, to check the correctness of our result, we find that by replacing $\rho$ to be zero, we can recover back to the case for SGD exactly.

To calculate the eigenvalue of $\max_i \lambda_{\max}(H_i)$, we will need to calculate the average number of the data that align with each other as follows:

$$\max_i \lambda_{\max}((I + \frac{\rho}{\alpha}H)H_i) = \max_i \lambda_{\max}((I + \frac{\rho}{n\alpha}\sum_{i=1}^n \nabla f_w(x_i)\nabla f_w(x_i)^T)\nabla f_w(x_i)\nabla f_w(x_i)^T)$$

$$= \max_i ||\nabla f_w(x_i)||^2 + \frac{\rho}{\alpha}||\nabla f_w(x_i)||^4 \frac{1}{2^c}$$

$$= 2(d+1)^{\frac{1}{2}}(1 + \frac{\rho}{\alpha}\frac{1}{2^c}2(d+1)^{\frac{1}{2}})$$

$$\tag{79}$$

$$\square$$

## C.13 Proof for theorem 3.3

*Proof.* We follow the construction of prior work Dexter et al. [2024] and focus on the term $E \operatorname{Tr}[\hat{J}^T \hat{J}]$ where $\hat{J} = I - \eta H_t - \frac{\eta\rho}{\alpha} H_t H$ (Note that $H_t = \sum_i x_i H_i$, where $x_i$ is Bernoulli with probability $\frac{B}{n}$ being 1) with the probability of sampling each sample being independent Bernoulli distribution. The construction of set $\{H_i\}_{i\in[n]}$ is such that $H_i = m e_1 e_1^T$, $\forall i \in [\sigma]$ and $H_i = 0$ otherwise, and $m = \frac{\lambda_1 n}{\sigma}$ so that $\lambda_{\max}(H) = \frac{\sigma}{n} m = \lambda_1$. Note that $\lambda_{\max}(S) = m\sigma$ and $\max_i \lambda_{\max}(H_i) = m$ and the coherence measure is exactly $\sigma$. Also under this construction, $E[\operatorname{Tr}[\hat{J}_k^T ... \hat{J}_1^T \hat{J}_1 ... \hat{J}_k]] = \operatorname{Tr}[E[(\hat{J}_1^T \hat{J}_1)^{2k}]]$ as all matrix involved are commuting with i.i.d sampled.

We have following:

$$
\begin{aligned}
E[\hat{J}^T \hat{J}] &= E[(I - \eta H_t - \frac{\eta\rho}{\alpha} H H_t)(I - \eta H_t - \frac{\eta\rho}{\alpha} H_t H)] \\
&= E[I - 2\eta H_t - \frac{\eta\rho}{\alpha}(H H_t + H_t H) + \eta H_t^2 + \frac{\eta^2\rho}{\alpha}(H_t^2 H + H H_t^2) + \frac{\eta^2\rho^2}{\alpha^2} H H_t^2 H] \\
&= I - 2\eta H_t - 2\frac{\eta\rho}{\alpha} H^2 + \eta H_t^2 + 2\frac{\eta^2\rho}{\alpha} H^3 + \frac{\eta^2\rho^2}{\alpha^2} H^4 + \eta^2(\frac{1}{Bn} - \frac{1}{n^2}) \sum_i (I + \frac{\rho}{\alpha} H) H_i^2 (I + \frac{\rho}{\alpha} H)
\end{aligned}
$$
(80)

Now, we calculate $e_1^T E[\hat{J}^T \hat{J}] e_1$ ($e_1$ is the only direction that involve interaction of different samples) will give us

$$
e_1^T E[\hat{J}^T \hat{J}] e_1 = 1 - 2\eta\lambda_1 - 2\frac{\eta\rho}{\alpha}\lambda_1^2 + \eta\lambda_1^2 + 2\frac{\eta^2\rho}{\alpha}\lambda_1^3 + \frac{\eta^2\rho^2}{\alpha^2}\lambda_1^4 + \eta^2(\frac{1}{Bn} - \frac{1}{n^2})(1 + \frac{\rho}{\alpha}\lambda_1)^2 \frac{n^2\lambda_1^2}{\sigma}
$$
(81)

We need the term to be smaller than 1 to avoid growing infinitely

$$
1 - 2\eta\lambda_1 - 2\frac{\eta\rho}{\alpha}\lambda_1^2 + \eta\lambda_1^2 + 2\frac{\eta^2\rho}{\alpha}\lambda_1^3 + \frac{\eta^2\rho^2}{\alpha^2}\lambda_1^4 + \eta^2(\frac{1}{Bn} - \frac{1}{n^2})(1 + \frac{\rho}{\alpha}\lambda_1)^2 \frac{n^2\lambda_1^2}{\sigma} \le 1 \quad (82)
$$

We can rearrange and obtain the following:

$$
-2\eta\lambda_1(1 + \frac{\rho}{\alpha}\lambda_1) + \eta^2\lambda_1^2(1 + \frac{\rho}{\alpha}\lambda_1)^2 + \eta^2(\frac{n}{B} - 1)(1 + \frac{\rho}{\alpha}\lambda_1)^2 \frac{\lambda_1^2}{\sigma} \le 0 \quad (83)
$$

We find that we can divide the equation on both side by $\eta\lambda_1(1 + \frac{\rho}{\alpha}\lambda_1)$ and have

$$
-2 + \eta\lambda_1(1 + \frac{\rho}{\alpha}\lambda_1) + \eta(\frac{n}{B} - 1)(1 + \frac{\rho}{\alpha}\lambda_1)\frac{\lambda_1}{\sigma} \le 0 \quad (84)
$$

Now, we can rearrange and have the following:

$$
\eta\lambda_1(1 + \frac{\rho}{\alpha}\lambda_1)(\frac{\frac{n}{B} - 1}{\sigma} + 1) \le 2 \quad (85)
$$

$$
\frac{\eta\lambda_1}{\sigma}(1 + \frac{\rho}{\alpha}\lambda_1)(\frac{n}{B} - 1 + \sigma) \le 2 \quad (86)
$$

Finally, we will have

$$
\lambda_1(1 + \frac{\rho}{\alpha}\lambda_1) \le \frac{2\sigma}{\eta}(\frac{n}{B} - 1 + \sigma)^{-1} \quad (87)
$$

The additional term $(1 + \frac{\rho}{\alpha}\lambda_1)$ result from the SAM modified surface gives a more restricted learning rate choice compared to the SGD. We can also check the result by setting $\rho = 0$ and will find that it reduce to the original SGD criterion. $\square$

### C.14   Theorem from Dexter et al. [2024]

**Theorem 1**   Let $\{\hat{J}_i\}_{i\in\mathbf{N}}$ be a sequence of i.i.d copies of $\hat{J}$ defined in linearized SGD. Let $\{H_i\}_{i\in[n]}$ have coherence measure $\sigma$. If

$$\lambda_{\max}(H) \geq \frac{2}{\eta} \ \text{ or } \ \lambda_{\max}(H) \geq \frac{\sigma}{\eta}(\frac{n}{B}-1)^{-\frac{1}{2}}, \ \text{ then } \lim_{k\to\infty} E||\hat{J}_k...\hat{J}_1|| = \infty \qquad (88)$$

**Theorem 2**   For every choice of $\lambda_{\max} > 0, n \in \mathbf{N}, B \in [n], \eta > 0$ and $\sigma \in [n]$, that satisfies:

$$\lambda_{\max} < \frac{2\sigma}{\eta}(\sigma + \frac{n}{B} - 1)^{-1} \qquad (89)$$

There exists a set of PSD matrices $\{H_i\}_{i\in[n]}$ such that $\lambda_{\max}(H) = \lambda_{\max}$ and $\lim_{k\to\infty} E||\hat{J}_k...\hat{J}_1|| < n$.

**Lemma 4.1**   Let $\hat{J}_i$ be independent Jacobians of SGD dynamics,

(1) If

$$\lambda_{\max} \geq \frac{2}{\eta} \ \text{ or } \ \lim_{k\to\infty}(\frac{\eta^2}{nB} - \frac{\eta^2}{n^2}) \sum_{y_1...y_k=1}^{n} ||H_{y_k}...H_{y_1}||_F = \infty \qquad (90)$$

then $\lim_{k\to\infty} E||\hat{J}_k...\hat{J}_1||_F^2 = \infty$

(2) If, for some $\epsilon \in (0,1)$,

$$\frac{\epsilon}{\eta} < \lambda_i(H) < \frac{2-\epsilon}{\eta} \ \ \forall i \in [d] \ \text{ and} \frac{1}{\epsilon^k} \lim_{k\to\infty}(\frac{\eta^2}{nB} - \frac{\eta^2}{n^2}) \sum_{y_1...y_k=1}^{n} ||H_{y_k}...H_{y_1}||_F = 0 \qquad (91)$$

then $\lim_{k\to\infty} E||\hat{J}_k...\hat{J}_1||_F^2 = 0$

