# OpenReview forum: "A Unified Stability Analysis of SAM vs SGD: Role of Data Coherence and Emergence of Simplicity Bias"
_NeurIPS.cc/2025/Conference — NeurIPS 2025 poster_

### Official Review · Reviewer_e2Jp · 2025-07-01

**Clarity:** 3
**Significance:** 2
**Originality:** 3
**Rating:** 4
**Confidence:** 2

**Summary:**

This paper explores the dynamics of specific optimization algorithms (SGD, random perturbations of SGD, and SAM) and their perceived bias towards flatter or simpler solutions, especially in overparameterized regimes. The authors suggest a linear stability analysis framework as a tool to understand and quantify this behavior. Their approach relies on data and gradient similarity and attempts to explain why certain solutions are preferred for these optimization algos. The experiments are limited to two-layer ReLU networks.

**Questions:**

I don't have any major (decision altering) questions.

**Ethical Concerns:**

["NO or VERY MINOR ethics concerns only"]

**Limitations:**

yes

**Paper Formatting Concerns:**

the references are names, not numbered as I believe they should be. Also there is no neurips checklist in the main document

**Quality:**

3

**Strengths And Weaknesses:**

This is a well-written paper and addresses an interesting property that has been evidenced in neural network training, that

This is interesting work but I'm afraid I'm not an expert enough in this area to properly validate much of the analysis. The analysis is limited to three optimization algorithms (SGD, SGD with noise, and SAM) and the experiments focus only on two-layer ReLU networks. From my background, the development appears correct though and the analysis is useful. These factors limit my ability to classify this as a "5:Accept: ..high impact on at least one sub-area of AI or moderate-to-high impact on more than one area of AI" so I feel a rating of 4 is appropriate.

---

> ### Author Rebuttal · Authors · 2025-07-30
>
> We thank the reviewer for the response. Here we provide brief summary for our paper to help understanding it in more depth and we are glad to answer any question that the reviewer may have:
> This paper is for the first time studies what properties of data (and not just global properties of the function) ensure stochastic optimization algorithms like Stochastic Gradient Descent (SGD) and Sharpness-Aware Minimization (SAM) settle in or escape from local minima in complex NN training, and why some local minima they find generalize better than others.
>
> At the heart of our work is the idea of data coherence — a new way to measure how consistent the gradients (training signals) are across different data points. We show that when the gradients point in more similar directions (i.e., high coherence), optimization tends to be more stable, and the resulting models tend to generalize better. SAM is better at finding such minima, and SGD will `settle' in some incoherent minima that SAM will escape from. We analyze this behavior mathematically in two layer ReLU networks. Using this framework, we prove that stability during training is influenced by both the data and the algorithm, and we show that SAM encourages the algorithm to prefer more coherent, stable solutions than vanilla SGD.
>
> For the novelty of our theory, we summarize as follows:
>
> 1. Role of coherence in the optimization: We highlight the role of data coherence in shaping optimization dynamics, showing for the first time the importance of distribution of data itself (beyond its global properties such as Lipschitz-ness) and hot it actively interacts with the geometry of the loss landscape. In particular, coherent data distributions lead to more stable optimization trajectories, while datasets with many outliers (i.e., low coherence) can exhibit instability. This perspective helps bridge the gap between optimization theory and data.
>
> 2. New analysis for SAM: Our framework provides a novel analysis of Sharpness-Aware Minimization (SAM), showing that SAM implicitly penalizes incoherent solutions. This reveals that SAM not only promotes flat minima but also enforces stronger alignment between the solution and the underlying data, offering new insight into its generalization behavior.
>
> 3. We show that coherent solutions often correspond to simpler activation patterns, providing a functional interpretation of coherence in terms of network behavior. This connection offers a new lens to understand generalization, establishing a link amongst solution simplicity, data alignment, and internal network representations.
>
> To ground our theory in practice, we conduct experiments that:
>
> 1. Show coherence improves generalization and guarantee the stability of the solution. The new connection between data and optimization process that reveal the role of data in optimization.
>
> 2. Show that SAM leads to simpler (lower-rank) feature representations, indicating that coherence is reflected in the activation patterns of neural networks. This insight leverages the layered structure of deep networks and bridges theoretical analysis with the internal behavior of practical architectures.
>
> 3. Extend beyond toy settings to larger models and datasets like CIFAR-10 (see response for Response for Reivewer k5uF). Overall, our work contributes a new theoretical tool based on coherence and local stability to explain and potentially design better training algorithms that align with the structure of the data.

---

### Official Review · Reviewer_tCPY · 2025-07-03

**Clarity:** 4
**Significance:** 4
**Originality:** 3
**Rating:** 5
**Confidence:** 3

**Summary:**

This work analyzes two widely used algorithms: SGD and SAM, where the focus is on understanding two properties of the trained network parameters (local minima): the sharpness/flatness of the minima and its relation to generalization, and the tendency of SGD to find simpler minima especially in overparametrized regimes.

The main results of this paper provide evidence that flatness and simplicity (sometimes) come together, and that insisting on flatness (like SAM does) coms with an inherent bias for simplicity.

To this end the authors utilize a linear stability framework and the data coherence measure derived by Dexter et al. Their first result (Theorem 3.1) concerns SGD with random perturbation shows that adding random noise to SGD doers not alter which minima are stable, and furthermore that once a minimum is unstable, the noise causes the iterates to diverge at a faster rate.

Their second result (that I found more interesting) shows that under quadratic loss, SAM dynamics are biased towards coherence measure that captures how sample gradients align, but SAM also penalizes directions with large curvature. In addition, they show a matching lower bound on the SAM divergence.

Finally, the authors consider realistic model architectures and quantify the coherence structure of different solutions, showing how this governs their stability under SGD and SAM.

**Questions:**

I found Definition 2 of $(C,r)$-generalizing solution a bit ill-defined and unnatural. What is a (C,r) generalizing solution? You define $W_1$ and $W_2$, but do not state what a $(C,r)$ generalizing solution is.

**Ethical Concerns:**

["NO or VERY MINOR ethics concerns only"]

**Final Justification:**

I think this paper should be presented at NeurIPS, as it tackles important issues.

**Limitations:**

Yes

**Quality:**

3

**Strengths And Weaknesses:**

Strengths: The considerations of this paper shed light on interesting properties of SGD and SAM, and address important phenomenon. I was able to verify most of the theoretical results except Theorem 3.5 and they appear correct. I believe this paper will be of broad interest to the NeurIPS community and vote acceptance.

Weaknesses: First, I believe related work should appear in the main body of the paper, not in an appendix.
Next, all of the analysis is based on behavior near local minima, and hence the linear approximation of the loss landscape is easier to handle. Finally, the paper only studies two-layer ReLU networks. In terms of novelty, the work appears to be bordering on being incremental, but I feel there is sufficient novelty in the SAM and the 2-layer results to warrant acceptance.

---

> ### Author Rebuttal · Authors · 2025-07-30
>
> 1. Weaknesses: First, I believe related work should appear in the main body of the paper, not in an appendix. Next, all of the analysis is based on behavior near local minima, and hence the linear approximation of the loss landscape is easier to handle. Finally, the paper only studies two-layer ReLU networks. In terms of novelty, the work appears to be bordering on being incremental, but I feel there is sufficient novelty in the SAM and the 2-layer results to warrant acceptance.
>
> We thank the reviewer for the thoughtful comments and for recognizing the novelty in our results. We have tried to provide all contextualized related works and cited them inline in the introduction itself. As such, we did not feel a separate related works section (which is mostly 'additiona related works') added a lot of value over what is already in the intro. If the reviewer strongly feels otherwise, we are happy to move it to the main paper. We agree that it plays an important role in framing the contributions of the paper.
>
> Regarding novelty, our coherence-based analysis offers a new perspective on how property of data interacts with optimization. Unlike prior works that focus solely on sharpness, we show that alignment in the loss landscape—driven by data coherence—can act as a stabilizing factor, revealing a previously underexplored mechanism for generalization. This insight leads to a theoretical breakthrough by integrating data-dependent property into the analysis of optimization and generalization. Furthermore, our framework bridges loss landscape geometry, distribution of data, and simplicity bias in neural networks. We view this as a key advancement, as simplicity bias is a defining feature of modern deep networks and often touted as an important aspect of explaining generalization (see ref[1][2]). Our results thus help connect optimization theory with broader aspects of deep learning, filling a critical gap in existing understanding.
>
> On the focus on local analysis, we fully acknowledge this as a limitation. Any approximation of neural networks for theoretical tractability, including neural tangent kernels,  infinite widths of intermediate layers, gradient flows, bayesian perspective on sgd, make some assumptions that may not always hold. However, we believe the characterization of behavior around local minima is still highly relevant in modern deep learning and is grounded in empirical observations. Training dynamics often enter a hovering regime where the loss plateaus and the optimizer oscillates near a minimum for a substantial portion of training. This regime is known to influence generalization and stability properties. Our analysis is intended as a first step to provide a rigorous foundation for understanding this important phase of training. Lastly, while it is true that our theoretical analysis is grounded in two-layer ReLU networks, we emphasize that this model class has been widely used to uncover foundational insights in deep learning theory. (see ref [3][4]). The simplifications being made for RELU constructions are highly relevant to modern data and neural network interactions. Moreover, we extend our experimental validation to deeper architectures (e.g., ResNet-18 on CIFAR-10) and show the correspondence to coherence and feature-rank trends unearthed through our theoretical analysis, suggesting that the insights transfer beyond the two-layer setting. We agree that generalizing the theory to deeper or more complex networks is an exciting and important future direction, but we hope the reviewer sees the importance of our work as a stepping stone in that direction. We appreciate the reviewer’s overall positive assessment and constructive feedback.
>
> [1] K. Gatmiry, Z. Li, S. J. Reddi, and S. Jegelka, “Simplicity bias via global convergence of sharpness minimization,” arXiv, 2024
>
> [2] Springer, J. M., Nagarajan, V., and Raghunathan, A. (2024). Sharpness-aware minimization
> enhances feature quality via balanced learning. ICLR.
>
> [3] El Mehdi Achour., The loss landscape of deep linear neural networks: a second-order analysis. JMLR 2024.
>
> [4] Kaiyue Wen. How does sharpness-aware minimization minimize sharpness? arXiv 2022.
>
>
>
> 2. I found Definition 2 of -generalizing solution a bit ill-defined and unnatural. What is a (C,r) generalizing solution? You define and , but do not state what a generalizing solution is.
>
> We thank the reviewer for pointing this out and apologize for the lack of clarity. In our work, a generalizing solution refers to a solution (i.e., a set of network weights) that achieves zero test classification error under the data distribution we mentioned in the main context (section 3.3). (i.e., $y = f_w(x) \;\;\forall x,y \sim \mathcal{D}$) To be more specific, we provide one example in the appendix C.1. To provide intuitive understanding, the C in (C,r) solution controls how many neurons are activated in the first layer, and r control the the flatness in the solution as mentioned in the Definition 2 and the paragraph underneath. We will revise the main text to more clearly introduce what we mean by a generalizing solution before presenting Definition 2. If the reviewer has a specific concern about the formulation, we would be happy to address it in greater detail.
>
> Note that the generalizing solutions encompass exponentially many parameter configurations due to inherent symmetries in neural networks such as hidden unit permutations and layer-wise scaling invariance. These transformations preserve important properties such as loss, gradient, and Hessian, resulting in a broad equivalence class of solutions with indistinguishable local geometry. Our work provides a principled characterization of the optimization dynamics within this rich class. Moreover, it demonstrates the compatibility of our framework with broader aspects of deep learning, including simplicity bias and representation learning.

---

> > ### Comment · Reviewer_tCPY · 2025-08-07
> >
> > I thank the authors for their response. After reading the rebuttal and other reviews, I still vote for accepting this submission due to the scope of the issues discussed herein, I would like to see this presented at NeurIPS, and keep my original rating.

---

### Official Review · Reviewer_fbHT · 2025-07-03

**Clarity:** 3
**Significance:** 2
**Originality:** 2
**Rating:** 5
**Confidence:** 4

**Summary:**

This paper studies the dynamical stability of a randomly perturbed version of SGD and SAM, owing to both algorithms' generalization capabilities. Framed in the setup of Dexter et al. (2024)'s coherence measure, the paper shows that SAM enforces a tighter coherence matrix, and connects this to work showing its better generalization capabilities compared to SGD. Experiments with a synthetic dataset are shown.

**Questions:**

* Could you better contextualize the novelty of this paper and distinguish it from Wu & Su (2023)? I'm not convinced by the appendix's statement that the MSE loss being used is a limitation of that work that is addressed here.

**Ethical Concerns:**

["NO or VERY MINOR ethics concerns only"]

**Final Justification:**

* My primary concern with the paper was regarding its novelty wrt prior work such as Wu & Su (2023). In their rebuttal, the authors clearly point out where that paper makes specific assumptions that this work does not--this clearly addresses my concern and demonstrates the novelty of this paper.
* The authors also address my other concerns clearly. The authors should add the discussion from their rebuttal to the camera-ready version (e.g., the distinction from prior work, which can be added to the appendix; notation issues).

Although I am deciding between a 4 and a 5, I do not have any further concerns with this paper, and therefore I am updating my score to 5.

**Limitations:**

yes

**Quality:**

3

**Strengths And Weaknesses:**

**Strengths:**

* This paper addresses an important topic (flat minima/generalization), and adds to a growing body of work that studies it through the lens of stability.
* The paper is generally well-written and easy to follow.
* Most of the relevant work in the area has been thoroughly discussed in the appendix.

**Weaknesses:**

* The novelty of this paper with respect to other work on dynamical stability (such as Wu & Su (2023)) is unclear. In Appendix B, the authors attempt to distinguish their work as, "... but compared to prior works [Wu et al., 2018, 2022, Wu and Su, 2023], instead of assuming mean square loss, we have more general abstraction to include different kind of loss function." This is not true: for example, Wu & Su (2023) use a generic loss function and Hessian throughout their analysis as well (similar to this paper), and only use the MSE as an example in Section 2. The rest of their work studies a general ERM problem, with a gradient $g$ and a Hessian $H$ (though their paper uses the associate empirical Fisher matrix $G$ instead of $H$).
* This paper would benefit from a brief explanation of Dexter et al. (2024), since that work is the basis for some of the theory and the experimental setup. As such, it is useful to readers to understand why the specific experimental setup is chosen.
* After Theorem 3.1, the paper states, "Further, in the stable regime, the iterates do not converge exactly to the minimum, but instead remain in a bounded region around it (part 3)". Is this true for all minima, or only when $w^* = 0$? To me, part 3 seems to instead suggest that while the iterates remain in a bounded region, that region may not be symmetric around the minimum; is this accurate?
* In Definition 2, it would be beneficial to use a consistent notation, i.e., either $W_{1, j}$ or $W_1[j]$, as used for $W_2$.
* In Appendix C.1, "Take the trace will be ..." --> "Then the trace is ..."
* In Appendix C.1, $e_1$ is not defined.

Wu, L., & Su, W. J. (2023). The Implicit Regularization of Dynamical Stability in Stochastic Gradient Descent. International Conference on Machine Learning. http://arxiv.org/abs/2305.17490

---

> ### Author Rebuttal · Authors · 2025-07-30
>
> 1. The novelty of this paper with respect to other work on dynamical stability (such as Wu \& Su (2023)) is unclear. In Appendix B, the authors attempt to distinguish their work as, "... but compared to prior works [Wu et al., 2018, 2022, Wu \& Su, 2023], instead of assuming mean square loss, we have more general abstraction to include different kind of loss function." This is not true: for example, Wu \& Su (2023) use a generic loss function and Hessian throughout their analysis as well (similar to this paper), and only use the MSE as an example in Section 2. The rest of their work studies a general ERM problem, with a gradient and a Hessian (though their paper uses the associate empirical Fisher matrix).
>
> We thank the reviewer for the helpful clarification. Upon revisiting Wu \& Su (2023), we agree that their broader framework is presented in terms of general empirical risk minimization, and that their analysis of generalization error and sharpness indeed applies to a general loss and Hessian (or Fisher). Our statement in Appendix B was too strong and will be revised accordingly. That said, we would like to clarify a subtle but important distinction: while their overall setup is general, the key linear stability analysis in Wu \& Su (2023) especially the parts relying on spectral properties of the Jacobian or stability matrix is carried out under the mean squared error (MSE) loss. The use of MSE enables concrete expressions for the Hessian and simplifications that arise from its symmetry and quadratic form. For instance, Proposition 3.2 builds on Equation (3), which assumes MSE, leading to the specific Hessian form in Equation (2). Proposition 3.4 similarly uses this structure, and its derivation in Equation (23), as noted in Appendix B.2, depends on Lemma 3.3 — both of which are MSE-specific. While Wu \& Su later discuss generalization using broader loss functions, their linear stability bounds are derived concretely under the MSE setting. We hope this clarifies the intended distinction, and we appreciate the opportunity to correct and clarify our prior statement.
>
> 2. This paper would benefit from a brief explanation of Dexter et al. (2024), since that work is the basis for some of the theory and the experimental setup. As such, it is useful to readers to understand why the specific experimental setup is chosen.
>
> We appreciate the reviewer for the question. Our work generalizes the stability framework of Dexter et al 2024 on SGD analysis to include Sharpness-Aware Minimization (SAM) and random perturbation SGD, revealing that SAM imposes a stricter and more geometry-sensitive stability condition than SGD. It proves that SAM preferentially selects solutions with higher inter-sample curvature alignment, even when sharpness is held constant. Unlike Dexter et al, which focuses solely on SGD, this paper formalizes the link between stability and simplicity bias across optimizers and instantiates the theory in two-layer ReLU networks, distinguishing generalizing from memorizing solutions by their coherence structure.
>
> 3. After Theorem 3.1, the paper states, "Further, in the stable regime, the iterates do not converge exactly to the minimum, but instead remain in a bounded region around it (part 3)". Is this true for all minima, or only when w*=0? To me, part 3 seems to instead suggest that while the iterates remain in a bounded region, that region may not be symmetric around the minimum; is this accurate?
>
> We thank the reviewer for the thoughtful question. We clarify that the statement following Theorem 3.1 holds more generally and is not restricted to the case where w*=0. Since we are interested in dyanamics around optima actual value of $w^*$ does not matter. The key point is that due to the presence of additive noise (independent of the curvature or symmetry of the loss landscape) iterates do not converge exactly to the minimizer, but instead fluctuate within a bounded region. If we look at equation (3), we can find that even under w*=0, there will still be non-zero update of the weight. This is not the result of the loss landscape but the noise that is added into the algorithm artificially. Here, we provide several random perturbation base algorithms (see ref [1][2]) that gradually decrease the noise during the training to achieve better convergence behavior as support for our statement.
>
> [1] Bisla., A. Low-pass filtering sgd for recovering flat optima in the deep learning optimization landscape. AISTATS, 2022.
>
> [2] Yong Liu. Random sharpness-aware minimization. NeurIPS, 2022.
>
> 4. In Definition 2, it would be beneficial to use a consistent notation.
>
> We thank the reviewer for addressing the issue. We will modify it accordingly in the updated version.
>
> 5. In Appendix C.1, $e_1$ is not defined.
>
> We appreciate reviewer for pointing that out. $e_1$ is the Canonical orthonormal basis with 1 as its first element and zero otherwise. We will incorporate this definition in the updated version.
>
> 6. Could you better contextualize the novelty of this paper and distinguish it from Wu Su (2023)? I'm not convinced by the appendix's statement that the MSE loss being used is a limitation of that work that is addressed here.
>
> We see our novelty in two different parts compared to prior work:
>
> First, Coherence base analysis: We use the Hessian coherence matrix as a central object to capture alignment between per-sample gradients. This sheds light on how SGD and SAM interact differently with different  data distributions — particularly in terms of stability and generalization — an aspect that is not a focus of Wu \& Su (2023). In contrast to their emphasis on the noise structure interaction with Fisher information matrix, our analysis reveals how data geometry influences the stability of solutions through coherence. This directly provides a theoretical bases for empirical observations on simplicity bias made by other papers. (see [1] [2])
>
> Second, bridging linear stability to neural network structure via activation patterns: A key novelty of our work is connecting linear stability theory to the activation patterns in ReLU networks. We show that generalizing solutions tend to induce more coherent gradients, which in turn correspond to simpler (i.e. more sparse, and more structured activation patterns. This provides a concrete way to interpret algorithmic bias (e.g., from SAM) in terms of neural architecture behavior — a perspective not developed in Wu \& Su or related stability works. We hope these clarifications help highlight how our work complements and extends the existing body of literature on optimization dynamics.
>
> [1] K. Gatmiry, Z. Li, S. J. Reddi, and S. Jegelka, “Simplicity bias via global convergence of sharpness minimization,” arXiv, 2024
>
> [2] Springer, J. M., Nagarajan, V., and Raghunathan, A. (2024). Sharpness-aware minimization
> enhances feature quality via balanced learning. ICLR.

---

> ### Comment · Reviewer_fbHT · 2025-08-04
>
> Thank you for the clear rebuttal. The authors have addressed my primary concern (novelty), citing specific parts of prior work that makes the assumptions that their paper claims. The rebuttal also clearly addresses the other issues I raised. I currently do not see any reason to reject this paper, so I will update my score to a 5.

---

### Official Review · Reviewer_k5uF · 2025-07-03

**Clarity:** 3
**Significance:** 2
**Originality:** 3
**Rating:** 4
**Confidence:** 3

**Summary:**

This paper presents a unified stability analysis for SGD and Sharpness-Aware Minimization. The authors develop a linear stability framework that covers standard SGD, a noise-injected variant, and SAM, centered on a data coherence matrix capturing the alignment of per-example Hessians. They prove that minima with high coherence remain stable under SGD while low-coherence (memorizing) solutions are unstable, explaining why SGD implicitly favors simple feature-sharing solutions. SAM further penalizes high curvature via an effective Hessian factor, imposing a stricter stability criterion that biases optimization toward even more coherent minima. Key contributions include exact stability conditions for both algorithms, matching lower bounds that demonstrate tightness, theoretical proofs linking coherence to an implicit simplicity bias, and an empirical study on a two-layer ReLU network showing that SAM consistently reduces coherence and effective feature rank relative to SGD.

**Questions:**

1. The analysis highly depends on several simplification assumptions, including local linearization, $(C,r)$-generalizing solutions, etc. In addition, the analysis stops at focus on a two-layer ReLU model. Are these main assumptions and conditions reasonable to assume and commonly used in real training of modern models? And is it possible to generalize the analysis to more complicated situations?  How do the authors anticipate these insights extending to deeper or more complex neural architectures? For example, would a deep convolutional or transformer model exhibit a similar coherence-driven stability behavior? Are there new challenges (e.g. multiple dominant Hessian directions, non-linear interactions between layers) in applying the coherence measure and stability criteria to much deeper networks?

2. Measuring and Using Coherence in Practice: The paper introduces the data coherence matrix as a key quantity. Is it feasible to compute or approximate this coherence measure for practical-scale datasets and networks, and could it be used as a diagnostic tool?

3. In the conclusion the authors suggest that their perspective could “open avenues for designing optimizers that align algorithmic bias with data geometry”, is there any insight or intuition on such algorithm based on the paper's discovery? Did the authors consider any concrete strategies, such as adaptively adjusting the learning rate or batch selection based on coherence, or designing new regularization terms that promote alignment of per-sample gradients?

4. As brought up by the authors in the limitation, would it be possible to provide more experiments with larger models and real-wold datasets for verification of the paper's findings?

**Ethical Concerns:**

["NO or VERY MINOR ethics concerns only"]

**Final Justification:**

The authors have addressed my concerns with thoughtful discussion and clear explanations. In particular, their clarification of the previously oversimplified theoretical settings enhances the paper’s core contributions and makes its central ideas more interesting and meaningful.

Accordingly, I am raising my score to 4. I also recommend that the authors incorporate key explanations and discussions from the rebuttal into the paper in the next revision.

**Limitations:**

The paper includes a separate paragraph on the limitations of the paper, in which they discuss the linear approximation and lack of more experiments. Other simplifications and possible practical implications can also be noted.

**Paper Formatting Concerns:**

No formatting concerns.

**Quality:**

3

**Strengths And Weaknesses:**

**Strength:**

 - Novel Unified Theory: The authors propose a unified linear stability framework that links SGD’s flat-minima bias and its implicit simplicity bias as arising from a common coherence-based mechanism. They derive exact stability conditions for SGD (with and without noise) and SAM in terms of a data-dependent coherence spectrum, establish matching lower bounds to confirm tightness, and formalize these insights as theorems extending prior analyses.

 - Empirical Validation: The authors include a well-designed empirical verification on a two layer ReLU network with analytically constructed minima from memorizing to simple solutions. They show that both SGD and SAM converge to low complexity solutions and that SAM achieves lower coherence and feature rank than SGD. These results, though based on a small model, confirm the theoretical predictions.
---
**Weakness:**

 - Reliance on Local Linearization: A core limitation of the analysis is its reliance on a linear approximation of the dynamics around a critical point (local minimum). All stability conclusions are derived in this linearized regime, assuming the optimizer is near a candidate solution and the loss can be approximated quadratically. This means the theory primarily addresses local stability (or instability) of a given minimum rather than the full global training trajectory. Real training often explores highly non-linear regions far from any critical point, and the local results may not extend to the full trajectory. This point is mentioned as limitation by the authors.

 - Strong Assumptions and Simplifications: Along with linearization, the analysis involves other simplifying assumptions. For example, theoretical results assume i.i.d. sampled data and focus on one specific model architecture for derivations (two-layer networks with ReLU). The constructed “$(C, r)$-generalizing solutions” framework (Definition 2) is clever for analysis, but it may not cover the full richness of real neural network solutions. Moreover, the study considers only vanilla SGD (with or without isotropic noise) and SAM. It’s unclear if and how the conclusions would change under some common training variations, including momentum and adaptive methods.

 - Unclear Direct Practical Implications: While the paper provides an insightful explanation for why SAM and SGD find certain minima, some of the innovations seems to be difficult for applications. For example, the coherence measure, which is a very important principle in the paper, is not something typically monitored during training, and computing per-example Hessian information is prohibitively expensive for large datasets. Thus, applying the coherence metric as a diagnostic or design principle in real training could be challenging.

 - Limited Empirical Scope: The experimental validation, while useful, is restricted to a toy setting (a two-layer ReLU network on a synthetic task with $n=100$ samples). This leaves open the question of how well the insights translate to larger-scale deep learning tasks and architectures. For instance, do modern deep networks exhibit the same coherence dynamics, and does SAM similarly reduce the effective feature rank in practice? Extending the experiments to more realistic settings would strengthen confidence that the coherence measure and stability conditions are broadly applicable.

---

> ### Author Rebuttal · Authors · 2025-07-30
>
> 1. Reliance on Local Linearization: The reviewer is concerned that the paper’s theoretical analysis is limited to a linear approximation near local minima, which may not capture the full dynamics of training. They question the broader applicability of the results, given that real training often explores non-linear regions far from any critical point.
>
> We thank the reviewer for highlighting this limitation, which we acknowledge in the main text. While our analysis is local, this regime remains highly relevant: empirical studies show that loss landscapes are often locally approximately quadratic, even in overparameterized models [1], motivating many theoretical frameworks for optimization and generalization [2, 3]. Since training often proceeds through plateau phases near minima, local dynamics are central to understanding stability. Like gradient flow and neural tangent kernel analyses, our work adopts simplifying assumptions that, while not universally valid, yield valuable insights. Our approach aligns with prior work (e.g., Dexter et al., 2024; Wu et al., 2022) demonstrating the utility of local approximations, and we will clarify these motivations in the revision.
>
> [1] Hao Li., Visualizing the loss landscape of neural nets. NeurIPS,2018
>
> [2] El Mehdi Achour., The loss landscape of deep linear neural networks: a second-order analysis. JMLR 2024.
>
> [3] Kaiyue Wen. How does sharpness-aware minimization minimize sharpness? arXiv 2022.
>
>
> 2. Strong Assumptions and Simplifications: The reviewer notes that the analysis relies on strong simplifying assumptions, such as i.i.d. data, two-layer ReLU networks, and basic optimizers (SGD, SAM). They question whether the proposed framework of generalizing solutions captures the complexity of real neural networks, and whether the conclusions extend to other training settings like momentum or adaptive methods.
>
> First, the i.i.d. sampling assumption is standard in theoretical deep learning (e.g., generalization bounds, stochastic processes) and reflects how minibatches are typically drawn in practice. Studying non-i.i.d. settings like curriculum or imbalanced sampling is a valuable direction for future work.
>
> Second, while $({C}, r)$-generalizing solutions are restricted, they reflect key structures observed in deep networks—especially in activation patterns. ReLU networks exhibit rich symmetries (e.g., permutation, rescaling) that yield exponentially many equivalent solutions with identical loss, gradient, and Hessian, making our framework practically relevant despite its constraints. Such simplifications are common to enable tractable analysis.
>
> Lastly, we focus on plain SGD and SAM to isolate the roles of noise and sharpness in a clean setting. Extending to optimizers like momentum SGD or Adam, which interact with curvature in subtle ways, is a promising direction we plan to pursue. We appreciate the reviewer’s suggestion and will note this in the revision.
>
> 3. Unclear Direct Practical Implications: The reviewer raises concern about the practicality of the proposed coherence measure, noting that it is computationally expensive and not typically used in training. They question how feasible it is to apply this concept in real-world settings as a diagnostic or design tool.
>
> Our work is intended as a first step in highlighting coherence as a potentially valuable conceptual lens for understanding training dynamics and generalization. While the coherence measure is not yet a standard diagnostic tool, this is also true for many theoretical quantities when first introduced. A relevant example is sharpness (e.g., maximum eigenvalue of the Hessian), which was originally difficult to compute at scale, yet it inspired successful optimization strategies such as SAM that approximate the principle indirectly. Similarly, although computing exact gradient coherence or per-sample Hessian quantities is currently expensive, we see this as a motivation for future work on scalable surrogates or proxies. Our contribution is to show that coherence connects to stability in a theoretically grounded way, suggesting that it could inform the design of future optimizers or monitoring tools, even if not computed directly. We will be glad to include this perspective and the challenges of practical deployment in the discussion section of the updated version.
>
> 4. Limited Empirical Scope : The reviewer questions the empirical scope of the paper, noting that experiments are limited to a small synthetic setup. Asking whether the theoretical insights, such as coherence dynamics and feature rank reduction by SAM, hold for large-scale models and datasets, and suggest extending the experiments to larger architectures to verify applicability.
>
> | Optimizer   | Rank               | Coherence        |
> |-------------|--------------------|------------------|
> | SGD         | 148.3333 ± 3.215   | 1.0045 ± 0.002    |
> | SAM 0.05    | 157.6667 ± 9.292   | 1.0052 ± 0.006    |
> | SAM 0.1     | 144.3333 ± 7.095   | 1.0771 ± 0.068    |
> | SAM 0.2     | 128.6667 ± 5.508   | 1.0907 ± 0.089    |
>
> We appreciate the reviewer’s suggestion. To investigate the scalability and practical relevance of our insights, we conducted additional experiments on the CIFAR-10 dataset using a ResNet-18 model (11.7M parameters). To approximate the coherence measure in a tractable way, we used the pairwise dot product of per-sample gradients normalized by the maximum gradient norm:  $\frac{\nabla l_i \nabla_jl}{\max_k ||\nabla l_k||}$ For further approximation, We compute this on a fixed subset of 100 samples (10 per class) to construct a $100 \times 100$ coherence matrix. For the feature rank calculation, we record the features from before the last linear layer for whole training dataset and use PCA to calculate the rank of feature with explain ratio up to 99.9 percent. According to our experiments, we find that the feature rank decrease as expected and the the approximated coherence measure also closely follow the training process despite milder trends due to the approximation.
>
> 5. Highly depends on several simplification assumptions: The analysis relies on simplifying assumptions (e.g., local linearization, two-layer ReLU models, specific generalizing solutions). The reviewer asks whether these assumptions are reasonable in modern training settings, and whether the results could extend to deeper architectures like CNNs or other modern structures. They also raise the question of potential new challenges in applying coherence-based stability analysis to such models (e.g., multiple dominant Hessian directions, non-linear layer interactions).?
>
> We appreciate the reviewer’s question. While our analysis relies on standard simplifying assumptions—local linearization and two-layer ReLU networks—these are common in the literature and have yielded useful insights into deep learning dynamics (e.g., [1][2]). Despite the simplified setting, our framework captures gradient coherence and stability, which we find to remain meaningful even in large-scale models like ResNet-18 (see CIFAR-10 results). Extending this theory to deeper architectures is a promising next step. Though deeper models introduce challenges like multi-modal Hessians and inter-layer interactions, approximations such as blockwise or layer-wise coherence may offer practical paths forward, which we plan to explore.
>
> [1] Lenaic Chizat and Francis Bach. Implicit bias of gradient descent for wide two-layer neural networks trained with the logistic loss. COLT, PMLR, 2020.
>
> [2] Lei Wu. Towards understanding generalization of deep learning: Perspective of loss landscapes. arXiv, 2017.
>
> 6. Measuring and Using Coherence in Practice: The paper proposes data coherence as a central concept. The reviewer asks whether this coherence measure can be feasibly computed or approximated at scale, and whether it could realistically serve as a diagnostic tool in practice.
>
> We appreciate the reviewer’s question. We agree that coherence is a promising quantity with the potential to provide deeper insight into training dynamics and generalization. While computing the full coherence matrix exactly is impractical for large-scale models, similar to how Hessian-related quantities (e.g., sharpness or trace) are often approximated, we believe coherence can be approximated efficiently using small batches or low-rank projections. As demonstrated in our experiments (e.g., ResNet18 on CIFAR-10), such approximations can already reveal meaningful trends. Developing principled and scalable approximations of coherence, and understanding how it correlates with training outcomes, is a promising direction for future work.
>
> 7. Potential direction for design of algorithm: The reviewer asks whether the paper offers any concrete ideas or intuitions for designing new optimizers that align algorithmic behavior with data geometry such as adjusting learning rates or batch selection based on coherence, or adding regularization terms tied to per sample gradient alignment.
>
> We thank the reviewer for the insightful suggestion. Our analysis aims to inspire optimizers that leverage gradient coherence to align with data geometry. One concrete idea is to adapt the learning rate based on coherence between mini-batches—reducing it when gradients are aligned, and increasing it when they diverge. Other possibilities include regularizers that promote gradient alignment or batch selection strategies favoring coherence. While we have not explored these experimentally, we view them as promising directions for future work.

---

> > ### Comment · Reviewer_k5uF · 2025-08-06
> >
> > Thank you for your thorough reply to my questions, and it address most of my concerns.
> >
> > My biggest concern was the over-simplification of the assumptions and model in the theory, and only verifications on the simple scenarios. The rebuttal of the authors basically answer the questions from all these perspectives.
> >
> > Meanwhile, I believe these discussions and experiments are important for better illustrating the story of the paper, so they should be incorporated into the main paper in the future.
> >
> > Overall, I will raise my score to 4.

---

### Note · Authors · 2025-08-14

We appreciate everyone for engaging in the discussion to improve the work. Our work studies the role of data relationships (via the data coherence metric) in the emergence and differentiation of simplicity bias in SAM vs SGD. All reviewers favored our theoretical contributions, and some also recognized our empirical evaluations (k5uF), theoretical contributions to this important topic (fbHT, e2Jp, tCPY), and the clarity of the paper (fbHT). Below, we provide a brief summary of reviewers' comments and our responses to help understand the overall scope of the rebuttal:
1. Reviewer k5uF:
The reviewer raised concerns about simplification, limited verification, and practical implications. We justified the simplification, highlighting that local behavior captures much of the training dynamics and is crucial for future optimizer development. To address the limited verification, we ran additional experiments with larger datasets and models, observing consistent behavior. Finally, we introduced new ideas based on our results, incorporating data's role in optimization, which can open up underexplored research directions. The reviewer agreed with our response and increased the score to 4.
2. Reviewer fbHT:
Reviewer fbHT raised concerns about the novelty of our work compared to Wu & Su (2023) and the assumptions in both works. We clarified these points and highlighted two key aspects of our work: coherence-based analysis and its connection to activation patterns in ReLU network. Our analysis shows how data geometry impacts solution stability, linking these insights to the simplicity bias through newly defined activation patterns. The reviewer agreed with our response and increased the score to 5.
3. Reviewer tCPY:
Reviewer tCPY major concerns were the novelty of the work vs existing works and the simple looking definition of the generalizing solution. In our response, we provided further clarification about our novelty, particularly the role of data in optimization and how they are related to the simplicity bias of the dynamics. Lastly, the reviewer agreed that the concerns were resolved and maintained the original score of 5.
4. Reviewer e2Jp:
Reviewer e2Jp did not have any major concerns that would change the decision. We provided a brief summary of our work to help the reviewer further understand the contributions. The reviewer claim insufficient expertise in the area and was not confident to give a rating of 5, so they maintained the rating of 4 as the result.

---

### Decision · Program_Chairs · 2025-09-17

**Decision:**

Accept (poster)

**Comment:**

This paper presents a unified stability analysis of SGD and SAM through the lens of data coherence, offering a novel perspective on the emergence of simplicity bias in optimization. The reviewers found the theoretical contributions to be interesting and technically solid, particularly the coherence-based framework that connects data geometry, stability of solutions, and generalization, as well as the new analysis of SAM’s role in preferring coherent minima. The main concerns centered on the simplifying assumptions (local linearization, two-layer ReLU networks, and basic optimizers) and the limited scope of empirical validation, which initially raised doubts about the generality and practical relevance of the results. However, the authors’ rebuttal was detailed and convincing: they clarified the novelty compared to prior work (notably Wu & Su, 2023), justified the theoretical assumptions, and supplemented the work with additional experiments on CIFAR-10/ResNet-18 showing that the coherence-based insights extend beyond toy settings. Reviewers engaged with the authors’ clarifications, and two reviewers raised their scores post-rebuttal, while the others maintained positive evaluations. Overall, the consensus is that despite its limitations, the work provides a principled and meaningful theoretical framework that enriches our understanding of optimizer bias, coherence, and generalization, making it a valuable contribution for the NeurIPS community. I recommend acceptance as a poster.